# Transposable elements regulate thymus development and function

**Jean-David Larouche[1,2], Céline M Laumont[3,4], Assya Trofimov[1,5,6,7], Krystel Vincent[1], Leslie Hesnard[1], Sylvie Brochu[1], Caroline Côté[1], Juliette F Humeau[1], Éric Bonneil[1], Joel Lanoix[1], Chantal Durette[1], Patrick Gendron[1], Jean-Philippe Laverdure[1], Ellen R Richie[8], Sébastien Lemieux[1,9], Pierre Thibault[1,10], Claude Perreault[1,2]***

[1]Institute for Research in Immunology and Cancer, Université de Montréal, Montreal, Canada; [2]Department of Medicine, Université de Montréal, Montréal, Canada; [3]Deeley Research Centre, BC Cancer, Victoria, Canada; [4]Department of Medical Genetics, University of British Columbia, Vancouver, Canada; [5]Department of Computer Science and Operations Research, Université de Montréal, Montréal, Canada; [6]Fred Hutchinson Cancer Center, Seattle, United States; [7]Department of Physics, University of Washington, Seattle, United States; [8]Department of Epigenetics and Molecular Carcinogenesis, University of Texas M.D. Anderson Cancer Center, Houston, United States; [9]Department of Biochemistry and Molecular Medicine, Université de Montréal, Montreal, Canada; [10]Department of Chemistry, Université de Montréal, Montréal, Canada

*For correspondence:
claude.perreault@umontreal.ca

Competing interest: The authors declare that no competing interests exist.

**Abstract** Transposable elements (TEs) are repetitive sequences representing ~45% of the human and mouse genomes and are highly expressed by medullary thymic epithelial cells (mTECs). In this study, we investigated the role of TEs on T-cell development in the thymus. We performed multiomic analyses of TEs in human and mouse thymic cells to elucidate their role in T-cell development. We report that TE expression in the human thymus is high and shows extensive age- and cell lineage-related variations. TE expression correlates with multiple transcription factors in all cell types of the human thymus. Two cell types express particularly broad TE repertoires: mTECs and plasmacytoid dendritic cells (pDCs). In mTECs, transcriptomic data suggest that TEs interact with transcription factors essential for mTEC development and function (e.g., PAX1 and REL), and immunopeptidomic data showed that TEs generate MHC-I-associated peptides implicated in thymocyte education. Notably, AIRE, FEZF2, and CHD4 regulate small yet non-redundant sets of TEs in murine mTECs. Human thymic pDCs homogenously express large numbers of TEs that likely form dsRNA, which can activate innate immune receptors, potentially explaining why thymic pDCs constitutively secrete IFN α/β. This study highlights the diversity of interactions between TEs and the adaptive immune system. TEs are genetic parasites, and the two thymic cell types most affected by TEs (mTECs and pDCs) are essential to establishing central T-cell tolerance. Therefore, we propose that orchestrating TE expression in thymic cells is critical to prevent autoimmunity in vertebrates.

## eLife assessment

This **important** study shows, based on analyses of single-cell RNA-seq data sets of thymus cells, that transposable elements (TEs) are broadly expressed in thymic stromal cells, especially in medullary thymic epithelial cells and plasmacytoid dendritic cells. The authors also show that at least some TE-derived peptides are presented by MHC-I molecules in the thymus. The study provides **solid** findings supporting a role of TEs in thymic T-cell selection and immune self-tolerance.

## Introduction

Self/non-self discrimination is a fundamental requirement of life (*Boehm, 2012*). In jawed vertebrates, the thymus is the only site where T lymphocytes can be properly educated to distinguish self from non-self (*Boehm and Swann, 2014*; *Suo et al., 2022*). This is vividly illustrated by Oncostatin M-transgenic mice, where T-cell production occurs exclusively in the lymph nodes (*Terra et al., 2005*). These mice harbor normal numbers of T-cell receptors (TCRs) αβ T cells but present severe autoimmunity and cannot fight infections (*Blais et al., 2008*). Intrathymic generation of a functional T-cell repertoire depends on choreographed interactions between the TCRs of thymocytes and peptides presented by major histocompatibility complex (MHC) molecules on various antigen-presenting cells (APCs) (*Zuñiga-Pflucker et al., 1989*). Positive selection depends on self-antigens presented by cortical thymic epithelial cells (cTECs) and ensures that TCRs recognize antigens in the context of the host's MHC molecules (*Breed et al., 2018*; *Dervović and Zúñiga-Pflücker, 2010*). The establishment of central tolerance depends on two main classes of APCs located in the thymic medulla: dendritic cells (DCs) and medullary TEC (mTEC) (*Lebel et al., 2020*; *Srinivasan et al., 2021*; *Cheng and Anderson, 2018*). Two other APC types have a more limited contribution to central tolerance: thymic fibroblasts and B cells (*Perera et al., 2016*; *Nitta et al., 2011*). High avidity interactions between thymic APCs and autoreactive thymocytes lead to thymocyte deletion (negative selection) or generation of regulatory T cells (Treg) (*Malhotra et al., 2016*).

The main drivers of central tolerance, mTECs and DCs, display considerable phenotypic and functional heterogeneity. Indeed, recent single-cell RNA-seq (scRNA-seq) studies have identified several subpopulations of mTECs: immature mTEC(I) that stimulate thymocyte migration to the medulla via chemokine secretion (*Lkhagvasuren et al., 2013*), mTEC(II) that express high levels of MHC and are essential to tolerance induction, fully differentiated corneocyte-like mTEC(III) that foster a pro-inflammatory microenvironment (*Laan et al., 2021*), and finally mimetic mTECs that express peripheral tissue antigens (*Michelson et al., 2022*). Three different proteins whose loss of function leads to severe autoimmunity, AIRE, FEZF2, and CHD4, have been shown to drive the expression of non-redundant sets of peripheral tissue antigens in mTECs (*Ramsey et al., 2002*; *Takaba et al., 2015*; *Tomofuji et al., 2020*). DCs, on the other hand, are separated into three main populations. Conventional DC 1 and 2 (cDC1 and cDC2) have an unmatched ability to present both endogenous antigens and exogenous antigens acquired via cross-presentation or cross-dressing (*Ginhoux et al., 2022*). Plasmacytoid DC (pDC) are less effective APCs than cDCs, their primary role being to produce interferon alpha (IFNα) (*Ginhoux et al., 2022*). Notably, thymic pDCs originate from intrathymic IRF8[hi] precursors, and, in contrast to extrathymic pDCs, they constitutively secrete high amounts of IFNα (*Colantonio et al., 2011*; *Lavaert et al., 2020*; *Le et al., 2020*). This constitutive IFNα secretion by thymic pDCs regulates the late stages of thymocyte development by promoting the generation of Tregs and innate CD8 T cells (*Xing et al., 2016*; *Hanabuchi et al., 2010*; *Martín Gayo et al., 2010*; *Martinet et al., 2015*; *Epeldegui et al., 2015*).

Transposable elements (TEs) are repetitive sequences representing ~45% of the human and mouse genomes (*Treangen and Salzberg, 2011*; *Deniz et al., 2019*). Most TEs can be grouped into three categories: the long and short interspersed nuclear elements (LINE and SINE, respectively) and the long terminal repeats (LTRs). These broad categories are subdivided into over 800 subfamilies based on sequence homology (*Bourque et al., 2018*). TE expression is typically repressed in host cells to prevent deleterious integrations of TE sequences in protein-coding genes (*Argueso et al., 2008*). Unexpectedly, TEs were recently found to be expressed at higher levels in human mTECs than in any other MHC-expressing tissues and organs (i.e., excluding the testis) (*Larouche et al., 2020*; *Carter et al., 2022*), suggesting a role for TEs in thymopoiesis. Since some TEs are translated and generate MHC I-associated peptides (MAP) (*Larouche et al., 2020*), they might induce TE-specific central tolerance (*Kassiotis, 2023*). Additionally, TEs provide binding sites to transcription factors (TFs) and stimulate cytokine secretion via the formation of double-stranded RNA (dsRNA) (*Chuong et al., 2016*; *Bogdan et al., 2020*; *Adoue et al., 2019*; *Lefkopoulos et al., 2020*; *Lima-Junior et al., 2021*). Hence, TEs could have pleiotropic effects on thymopoiesis. To evaluate the role of TEs in thymopoiesis, we adopted a multipronged strategy beginning with scRNA-seq of human thymi and culminating in MS analyses of the MAP repertoire of mouse mTECs.

## Results

### LINE, LTR, and SINE expression shows extensive variations during ontogeny of the human thymus

We first profiled TE expression in various thymic cell populations during development. To do so, we quantified the expression of 809 TE subfamilies (classified according to the RepeatMasker annotations) in the scRNA-seq dataset of human thymi created by *Park et al., 2020*. Cells were clustered in 19 populations representing the main constituents of the thymic hematolymphoid and stromal compartments (*Figure 1a*, *Figure 1—figure supplement 1*). The expression of TE subfamilies was quantified at all developmental stages available, ranging from 7 post-conception weeks (pcw) to 40 years of age (*Supplementary file 1a*). Unsupervised hierarchical clustering revealed three clusters of TE subfamilies based on their pattern of expression during thymic development (*Figure 1b*, upper panel): (i) maximal expression at early embryonic stages persisting, albeit at lower levels, throughout ontogeny (cluster 1), (ii) an expression specific to a given timepoint (cluster 2), or (iii) a high expression at early embryonic stages that decreases rapidly at later timepoints (cluster 3). LINE and SINE subfamilies were enriched in cluster 1, whereas LTR subfamilies were significantly enriched in clusters 2 and 3 (*Figure 1b*, lower panel). Expression of individual LINE and SINE subfamilies was highly shared among different cell types (*Figure 1d*). In contrast, the LTR subfamilies' expression pattern was shared by fewer cell subsets and adopted a quasi-random distribution (*Figure 1d*). The pattern of expression assigned to TE subfamilies (*Figure 1c*, innermost track) was not affected by the proportion of cells of different developmental stages (embryonic or postnatal) (*Figure 1c*, outermost track, and *Figure 1—figure supplement 2*). This suggests that our observations do not result from a bias in the composition of the dataset. To gain further insights into the expression of TE subfamilies, we studied two biological processes known to regulate TE expression in other contexts: cell proliferation and expression of KRAB zinc-finger proteins (KZFP) (*Brocks et al., 2018*; *Imbeault et al., 2017*). Cell cycling scores negatively correlated with TE expression in various thymic cell subsets, particularly for LINE and SINE subfamilies shared among cell types (*Figure 1—figure supplement 3* and *Supplementary file 1b*), whereas analysis of KZFP expression identified ZNF10 as a probable repressor of L1 subfamilies in Th17 and NK cells (*Figure 1—figure supplement 4* and *Supplementary file 1c*). Thus, we conclude that the expression of the three main classes of TEs shows major divergences as a function of age and thymic cell types.

### TEs form interactions with transcription factors regulating thymic development and function

TEs provide binding sites to TFs (*Chuong et al., 2016*; *Kunarso et al., 2010*; *Sundaram et al., 2014*), and T-cell development is driven by the coordinated timing of multiple changes in transcriptional regulators (*Hosokawa and Rothenberg, 2021*). We, therefore, investigated interactions between TE subfamilies and TFs during the development of the human thymus. Two criteria defined an interaction: (i) a significant and positive correlation between the expression of a TF and a TE subfamily in a given cell population, and (ii) the presence of the TF binding motif in the loci of the TE subfamily (*Figure 2a*). Additionally, we validated the correlations we obtained using a bootstrap procedure to ascertain their reproducibility (see 'Materials and methods' for details). This procedure removed weakly correlated TF-TE pairs (*Figure 2b*). TF-TE interactions were observed in all thymic cell populations (*Figure 2c and d*, *Figure 2—figure supplement 1*, and *Supplementary file 1d*). Numerous TF-TE interactions were conserved between hematolymphoid and stromal cell subsets (*Figure 2e*). However, the number of interactions and the complexity of the interaction networks were much higher in mTECs than in other cell populations (*Figure 2c and d*, *Figure 2—figure supplements 1* and *2*).

Several TFs instrumental in thymus development and thymopoiesis interacted with TE subfamilies (*Figure 2—figure supplement 2* and *Supplementary file 1d*). These TFs include the NFKB1 and REL subunits of the NF-κB complex and PAX1 in mTECs (*Baik et al., 2016*; *Akiyama et al., 2008*; *Yamazaki et al., 2020*) and JUND in thymocytes (*Meixner et al., 2004*). In DCs, the most notable TF-TE interactions involved interferon regulatory factors (IRF), which regulate the late stages of T-cell maturation, and TCF4, which is essential for pDC development (*Xing et al., 2016*; *Cisse et al., 2008*). This observation is consistent with evidence that TEs have shaped the evolution of IFN signaling networks (*Chuong et al., 2016*). Finally, we found significant interactions between CTCF and TE

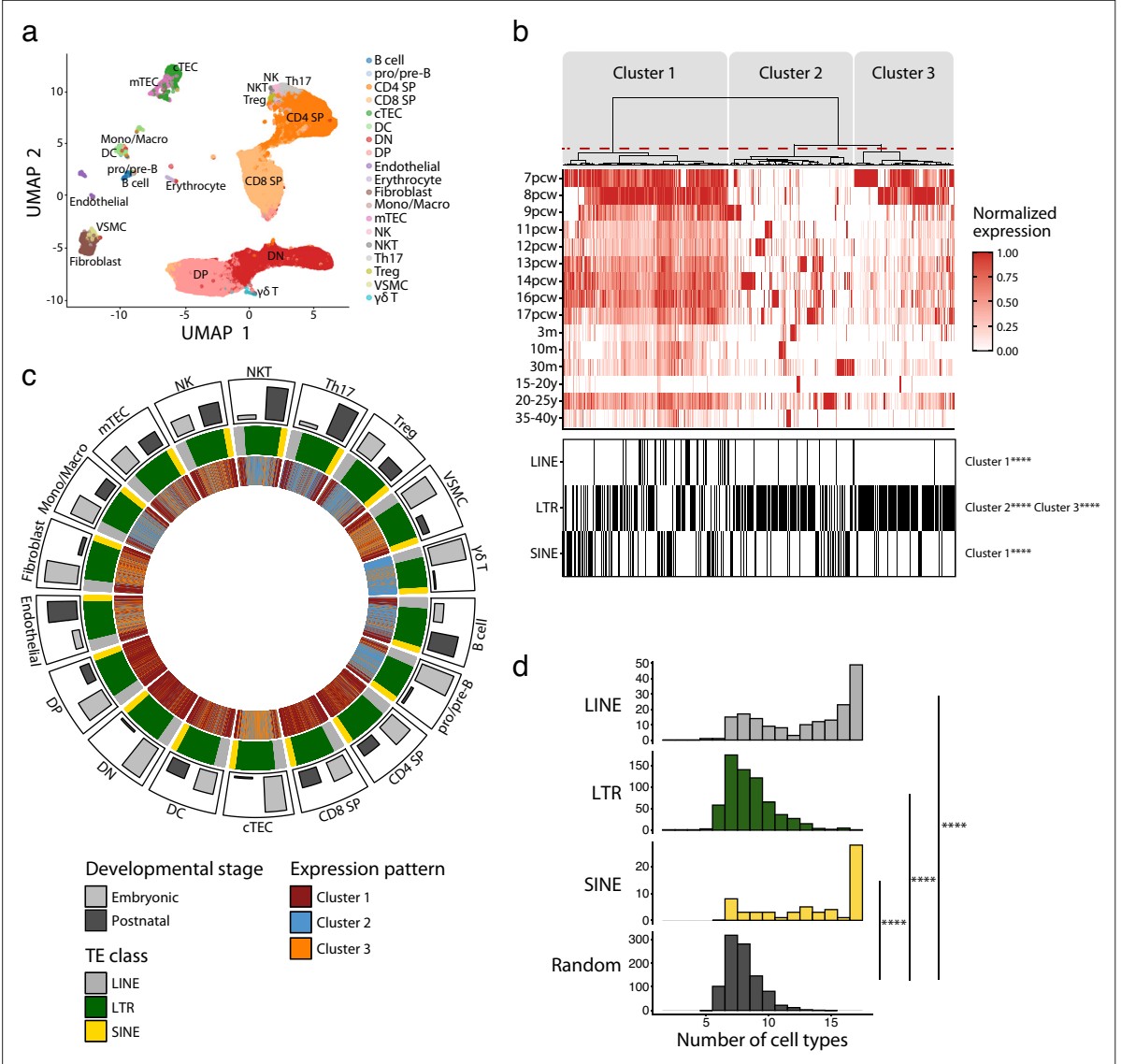

**Figure 1.** Long interspersed nuclear elements (LINE), short interspersed nuclear elements (SINE), and long terminal repeats (LTRs) exhibit distinct expression profiles in human thymic cell populations. (**a**) UMAP depicting the cell populations present in human thymi (CD4 SP, CD4 single positive thymocytes; CD8 SP, CD8 single positive thymocytes; cTEC, cortical thymic epithelial cells; DC, dendritic cells; DN, double negative thymocytes; DP, double positive thymocytes; Mono/Macro, monocytes and macrophages; mTEC, medullary thymic epithelial cells; NK, natural killer cells; NKT, natural killer T cells; pro/pre-B, pro-B and pre-B cells; Th17, T helper 17 cells; Treg, regulatory T cells; VSMC, vascular smooth muscle cell). Cells were clustered in 19 populations based on the expression of marker genes from *Lefkopoulos et al., 2020*. (**b**) Upper panel: heatmap of transposable element (TE) expression during thymic development, with each column representing the expression of one TE subfamily in one cell type. Unsupervised hierarchical clustering was performed, and the dendrogram was manually cut into three clusters (red dashed line). Lower panel: the class of TE subfamilies and significant enrichments in the three clusters (Fisher's exact tests; ****p≤0.0001). pcw, post-conception week; m, month; y, year. (**c**) Circos plot showing the expression pattern of TE subfamilies across thymic cells. From outermost to innermost tracks: (i) proportion of cells in embryonic and postnatal samples, (ii) class of TE subfamilies, and (iii) expression pattern of TE subfamilies identified in (**b**). TE subfamilies are in the same order for all cell types. (**d**) Histograms showing the number of cell types sharing the same expression pattern for a given TE subfamily. LINE (n = 171), LTR (n = 577), and SINE (n = 60) were compared to a randomly generated distribution (n = 809) (Kolmogorov–Smirnov tests, ****p≤0.0001).

The online version of this article includes the following figure supplement(s) for figure 1:

**Figure supplement 1.** Annotation of human thymic cell populations.

**Figure supplement 2.** Assignment to cluster 2 is independent of the developmental stage of cells.

**Figure supplement 3.** Transposable element (TE) expression is negatively correlated with cell proliferation.

**Figure supplement 4.** KRAB zinc-finger proteins (KZFP) repress transposable element (TE) expression in the hematopoietic lineage of the human thymus.

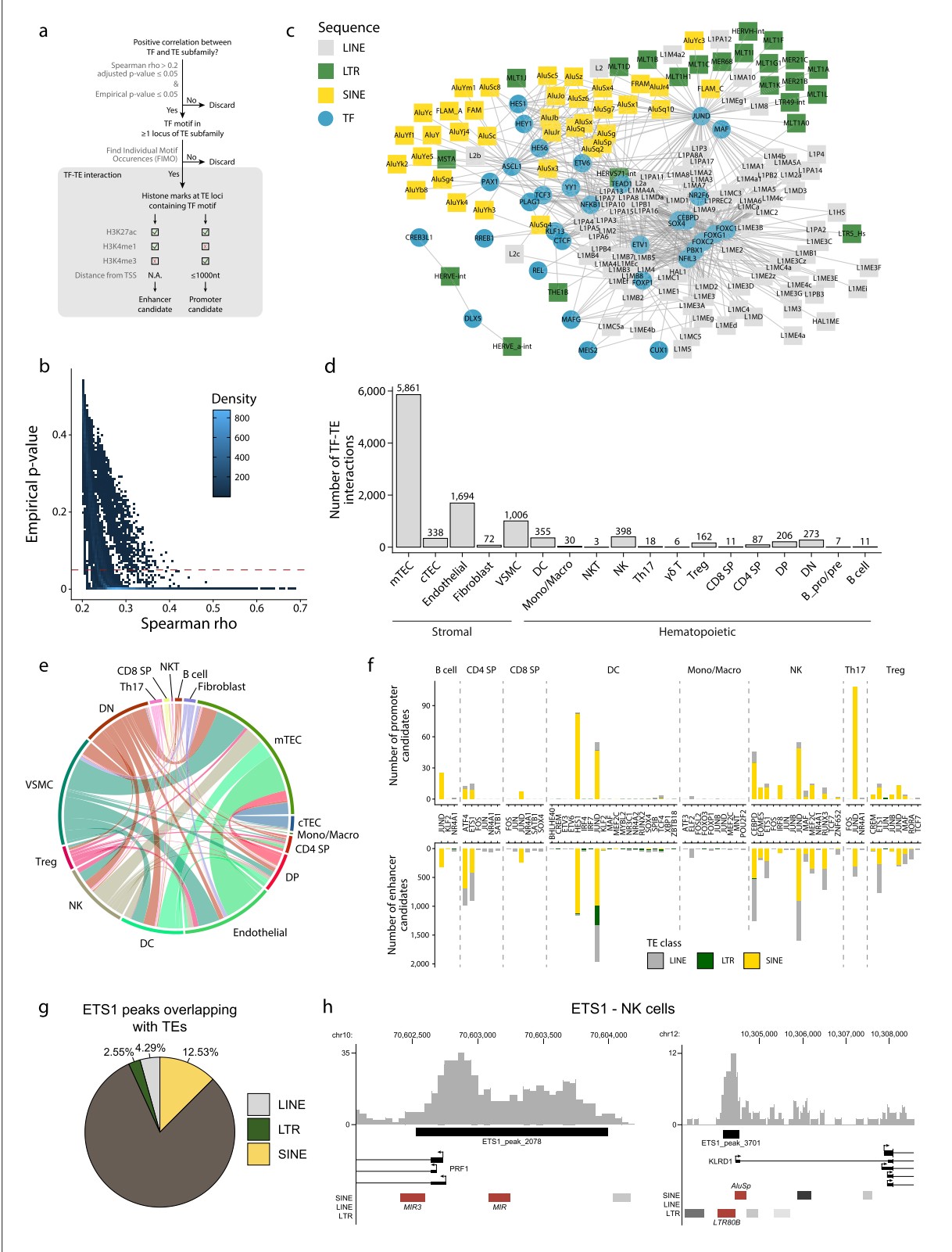

**Figure 2.** Transposable elements (TEs) shape complex gene regulatory networks in human thymic cells. (**a**) The flowchart depicts the decision tree for each TE promoter or enhancer candidate. (**b**) Density heatmap representing the correlation coefficient and the empirical p-value determined by bootstrap for transcription factor (TF) and TE pairs in each cell type of the dataset. The color code shows density (i.e., the occurrence of TF-TE pairs at a specific point). (**c**) Connectivity map of interactions between TEs and TFs in medullary thymic epithelial cells (mTECs). For visualization purposes, only

*Figure 2 continued on next page*

*Figure 2 continued*

TF-TE pairs with high positive correlations (Spearman correlation coefficient ≥ 0.3 and p-value adjusted for multiple comparisons with the Benjamini–Hochberg procedure ≤ 0.05) and TF binding sites in ≥1% of TE loci are shown. (**d**) Number of TF-TE interactions for each thymic cell population. (**e**) Sharing of TF-TE pairs between thymic cell types. (**f**) Number of promoter (top) or enhancer (bottom) TE candidates per TF in hematopoietic cells of the thymus. (**g**) The proportion of statistically significant peaks overlapping with TE sequences in ETS1 ChIP-seq data from NK cells. (**h**) Genomic tracks depicting the colocalization of ETS1 occupancy (i.e., read coverage) and TE sequences (in red) in the upstream region of two genes in ETS1 ChIP-seq data from NK cells. Statistically significant ETS1 peaks are indicated by the black rectangles.

The online version of this article includes the following figure supplement(s) for figure 2:

**Figure supplement 1.** Interaction networks between transcription factors (TFs) and transposable element (TE) subfamilies.

**Figure supplement 2.** Frequency of interactions between transcription factors (TFs) and transposable element (TE) subfamilies in thymic cells.

**Figure supplement 3.** Transposable element (TE) subfamilies occupying larger genomic spaces interact more frequently with transcription factor (TF).

subfamilies in mTECs and endothelial cells, suggesting that the binding of CTCF to TE sequences affects the tridimensional structure of the chromatin in the thymic stroma (*Choudhary et al., 2020*). Interestingly, LINE and SINE subfamilies that occupy more genomic space interacted with higher numbers of transcription factors (*Figure 2—figure supplement 3*).

Using data from the ENCODE consortium for hematopoietic cells (*Consortium, 2012*; *Luo et al., 2020*), we looked at the histone marks at the TE loci identified as TF interactors by our analyses (i.e., correlated with TF expression and containing the TF binding motif). The objective was to determine if they could act as promoters or enhancers (*Figure 2a* and *Supplementary file 1e*). We found several TE promoter and enhancer candidates in all eight hematopoietic cell types analyzed, with a striking overrepresentation of LINE and SINE compared to LTR sequences (*Figure 2f* and *Supplementary file 1f*). Finally, we analyzed publicly available ChIP-seq data of ETS1, an important TF for NK cell development (*Taveirne et al., 2020*), to confirm its ability to bind TE sequences. Indeed, 19% of ETS1 peaks overlap with TE sequences (*Figure 2g*). Notably, ETS1 peaks overlapped with TE sequences (*Figure 2h*, in red) in the promoter regions of PRF1 and KLRD1, two genes critical for NK cells' effector functions (*Kim et al., 2014*; *Gunturi et al., 2004*). Hence, our data suggest that TEs affect thymic development and function by providing binding sites to multiple TFs.

## TEs are highly and differentially expressed in human thymic APC subsets

We next sought to determine whether the high expression of TEs reported in mTECs (*Bourque et al., 2018*; *Argueso et al., 2008*) was limited to this cell subset or was found in other thymic cell types. Since several thymic stromal cells reach maturity after birth (*Bornstein et al., 2018*), we selected postnatal samples for the following analyses. We computed two distinct Shannon entropy indices: one for the global diversity of TEs expressed by all cells of a given population and another for the median value of TE diversity expressed by individual cells of a population (*Figure 3a*). Then, we computed a linear model to represent the diversity of TEs expressed by a cell population based on the diversity of TEs expressed by individual cells (*Figure 3a*, blue curve). Two salient findings emerged from this analysis. First, the diversity of TEs expressed in the T-cell lineage decreases during differentiation according to the following hierarchy: DN thymocytes > DP thymocytes > SP thymocytes (*Figure 3a*, *Figure 3—figure supplement 1*). Second, among the populations of thymic APCs implicated in positive and negative selection (*Figure 3a*, orange dots), cTECs, mTECs, and DCs expressed broader repertoires of TEs than B cells and fibroblasts. While cTECs and DCs expressed highly diverse TE repertoires at both the population and individual cell levels, the breadth of TE expression in mTECs was found only at the population level (*Figure 3a*). Accordingly, intercellular heterogeneity (i.e., deviation from the linear model) was higher for mTECs than other cell populations (*Figure 3b*).

We next focused on thymic APCs expressing the broadest TE repertoires: cTECs, mTECs, and DCs (*Figure 3a*). To this end, we annotated these APC subpopulations based on previously published lists of marker genes (*Figure 3c*, *Figure 3—figure supplement 2*; *Park et al., 2020*; *Bautista et al., 2021*). We performed differential expression analyses to determine whether some TE subfamilies were overexpressed in specific APC subsets. pDCs and mTEC(II) overexpressed a broader TE repertoire than other APCs: 32.01% of subfamilies were overexpressed in pDCs and 10.88% in mTEC(II) (*Figure 3d* and *Supplementary file 1g*). The nature of the overexpressed TEs differed between pDCs and other

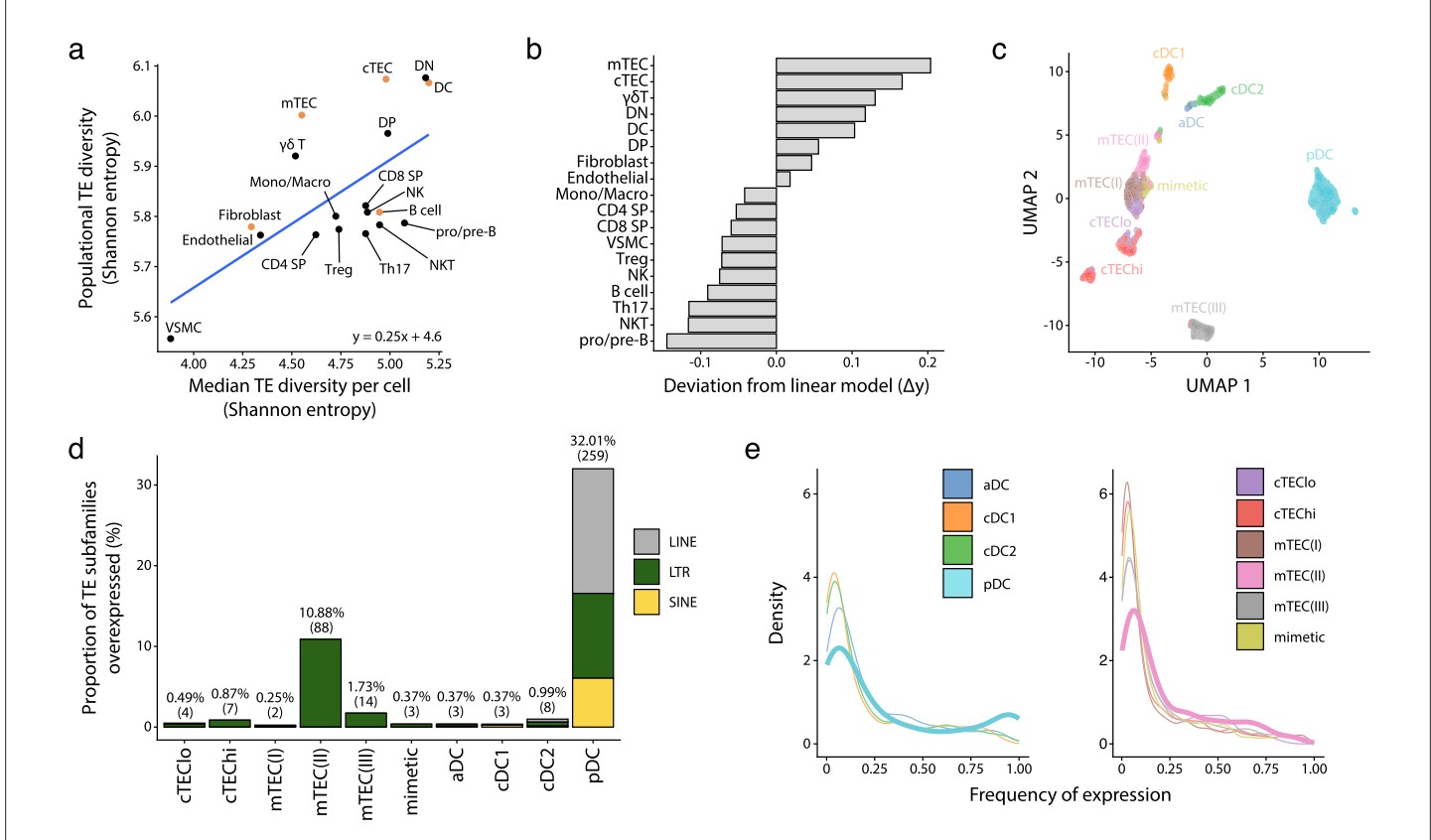

**Figure 3.** Human plasmacytoid dendritic cells (pDCs) and mTEC(II) express diverse and distinct repertoires of transposable element (TE) sequences. (**a**) Diversity of TEs expressed by thymic populations measured by Shannon entropy. The x and y axes represent the median diversity of TEs expressed by individual cells in a population and the global diversity of TEs expressed by an entire population, respectively. The equation and blue curve represent a linear model summarizing the data. Thymic antigen-presenting cell (APC) subsets are indicated in orange. (**b**) Difference between the observed diversity of TEs expressed by cell populations and the one expected by the linear model in (**A**). (**c**) UMAP showing the subsets of thymic APCs (aDC, activated DC; cDC1, conventional DC1; cDC2, conventional DC2). (**d**) Bar plot showing the number and class of differentially expressed TE subfamilies between APC subsets. (**e**) Frequency of expression of TE subfamilies by the different APC subsets. The distributions for pDCs and mTEC(II) are highlighted in bold. mTEC, medullary thymic epithelial cell.

The online version of this article includes the following figure supplement(s) for figure 3:

**Figure supplement 1.** Transposable element (TE) expression decreases during thymocyte differentiation.

**Figure supplement 2.** Annotation of human thymic antigen-presenting cell subsets.

**Figure supplement 3.** Differential transposable element (TE) expression in metacells of human thymic antigen-presenting cells.

thymic APC subsets. Indeed, pDCs overexpressed LTRs, LINEs, and SINEs, including several Alu and L1 subfamilies (***Figure 3d*** and ***Supplementary file 1g***). In contrast, other thymic APCs predominantly overexpressed LTRs.

TE expression showed wildly divergent levels of intercellular heterogeneity in APC subsets. Indeed, whereas most TE subfamilies were expressed by <25% of cells of the mTEC(II) population, an important proportion of TEs were expressed by >75% of pDCs (***Figure 3e***). To evaluate this question further, we compared TE expression between metacells of thymic APCs; metacells are small clusters of cells with highly similar transcription profiles. This analysis revealed that overexpression of TE subfamilies was shared between pDC metacells but not mTEC(II) metacells, reinforcing the idea that TE expression adopts a mosaic pattern in the mTEC(II) population (***Figure 3—figure supplement 3***). We conclude that cTECs, mTECs, and DCs express broad TE repertoires. However, two subpopulations of thymic APCs clearly stand out. pDCs express an extremely diversified repertoire of LTRs, SINEs, and LINEs, showing limited intercellular heterogeneity, whereas the mTEC(II) population shows a highly heterogeneous overexpression of LTR subfamilies.

## TE expression in human pDCs is associated with dsRNA structures

The high expression of a broad repertoire of TE sequences in thymic pDCs was unexpected (*Figure 3d*). LINE and SINE subfamilies, in particular, were highly and homogeneously expressed by thymic pDCs (*Figure 4a*). Constitutive IFNα secretion is a feature of thymic pDCs not found in extrathymic pDCs. We, therefore, hypothesized that this constitutive IFNα secretion by thymic pDCs might be mechanistically linked to their TE expression profile. We first assessed whether thymic and extrathymic pDCs have similar TE expression profiles by reanalyzing scRNA-seq data from human spleens published by *Madissoon et al., 2019*; *Figure 4—figure supplement 1a and b*. This revealed that extrathymic pDCs express TE sequences at similar or lower levels than other splenic cells (*Figure 4—figure supplement 1c and d*). We then used pseudobulk RNA-seq methods to perform a differential expression analysis of TE subfamilies between thymic and splenic pDCs. This analysis confirmed that TE expression was globally higher in thymic than in extrathymic pDCs (*Figure 4b*). Since TE overexpression can lead to the formation of dsRNA (*Lefkopoulos et al., 2020*; *Lima-Junior et al., 2021*), we investigated if such structures were found in thymic pDCs. pDCs were magnetically enriched from primary human thymi following labeling with anti-CD303 antibody (a marker of pDCs). Then, pDC-enriched thymic cells were stained with an antibody against CD123 (another marker of pDCs) and the J2 antibody that stains dsRNA. The intensity of the J2 signal was more than tenfold higher in CD123$^+$ relative to CD123$^-$ cells (*Figure 4c and d*). We conclude that thymic pDCs contain large amounts of dsRNAs. To evaluate if these dsRNAs arise from TE sequences, we analyzed in thymic APC subsets the proportion of the transcriptome assigned to two groups of genomic sequences known as important sources of dsRNAs: TEs and mitochondrial genes (*Sadeq et al., 2021*). Strikingly, whereas the percentage of reads from mitochondrial genes was typically lower in pDCs than in other thymic APCs, the proportion of the transcriptome originating from TEs was higher in pDCs (~22%) by several orders of magnitude (*Figure 4—figure supplement 2*). Finally, we performed gene set enrichment analyses to ascertain if the high expression of TEs by thymic pDCs was associated with specific gene signatures. These analyses highlighted signatures of antigen presentation, immune response, and interferon signaling in thymic pDCs (*Figure 4e* and *Supplementary file 1h*). Notably, thymic pDCs harbored moderate yet significant enrichment of gene signatures of RIG-I and MDA5-mediated IFN α/β signaling compared to all other thymic APCs (*Figure 4e* and *Supplementary file 1h*). Altogether, these data support a model in which the high and ubiquitous expression of TEs in thymic pDCs would lead to the formation of dsRNAs triggering innate immune sensors, which might explain their constitutive secretion of IFN α/β.

## AIRE, CHD4, and FEZF2 regulate distinct sets of TE sequences in murine mTECs

The essential role of mTECs in central tolerance hinges on their ability to ectopically express tissue-restricted genes, whose expression is otherwise limited to specific epithelial lineage (*Sansom et al., 2014*; *Pierre et al., 2017*). This promiscuous gene expression is driven by AIRE, CHD4, and FEZF2 (*Ramsey et al., 2002*; *Takaba et al., 2015*; *Tomofuji et al., 2020*). We, therefore, investigated the contribution of these three genes to the expression of TE subfamilies in the mTEC(II) population (*Figure 3d*). First, we validated that mTEC(II) express *AIRE, CHD4,* and *FEZF2* in the human scRNA-seq dataset (*Figure 5a*). Next, we analyzed published murine mTEC RNA-seq data to assess the regulation of TE sequences by AIRE, CHD4, and FEZF2. Differential expression analyses between knock-out (KO) and wild-type (WT) mice showed that these three factors regulate TE sequences, but the magnitude and directionality of this regulation differed (*Figure 5b* and *Supplementary file 1i*). Indeed, while CHD4 had the biggest impact on TE expression by inducing 433 TE loci and repressing 463, FEZF2's impact was minimal, with 97 TE loci induced and 60 repressed (*Figure 5b*). Besides, AIRE mainly acted as a repressor of TE sequences, with 326 loci repressed and 171 induced (*Figure 5b*). Interestingly, there was minimal overlap between the TE sequences regulated by AIRE, CHD4, and FEZF2, indicating that they have non-redundant roles in TE regulation (*Figure 5c*). Additionally, AIRE, CHD4, and FEZF2 preferentially targeted LTR and LINE elements, with significant enrichment of specific subfamilies such as MTA_Mm-int and RLTR4_Mm that are induced by Aire and Fezf2, respectively (*Figure 5d* and *Figure 5—figure supplement 1a*). While AIRE and CHD4 preferentially targeted evolutionary young TE sequences, the age of the TE sequence did not seem to affect the regulation by FEZF2 (*Figure 5—figure supplement 1b*). We also noticed that the distance between regulated TE loci was smaller than the distributions of randomly selected TEs (*Figure 5—figure supplement 1c*). This

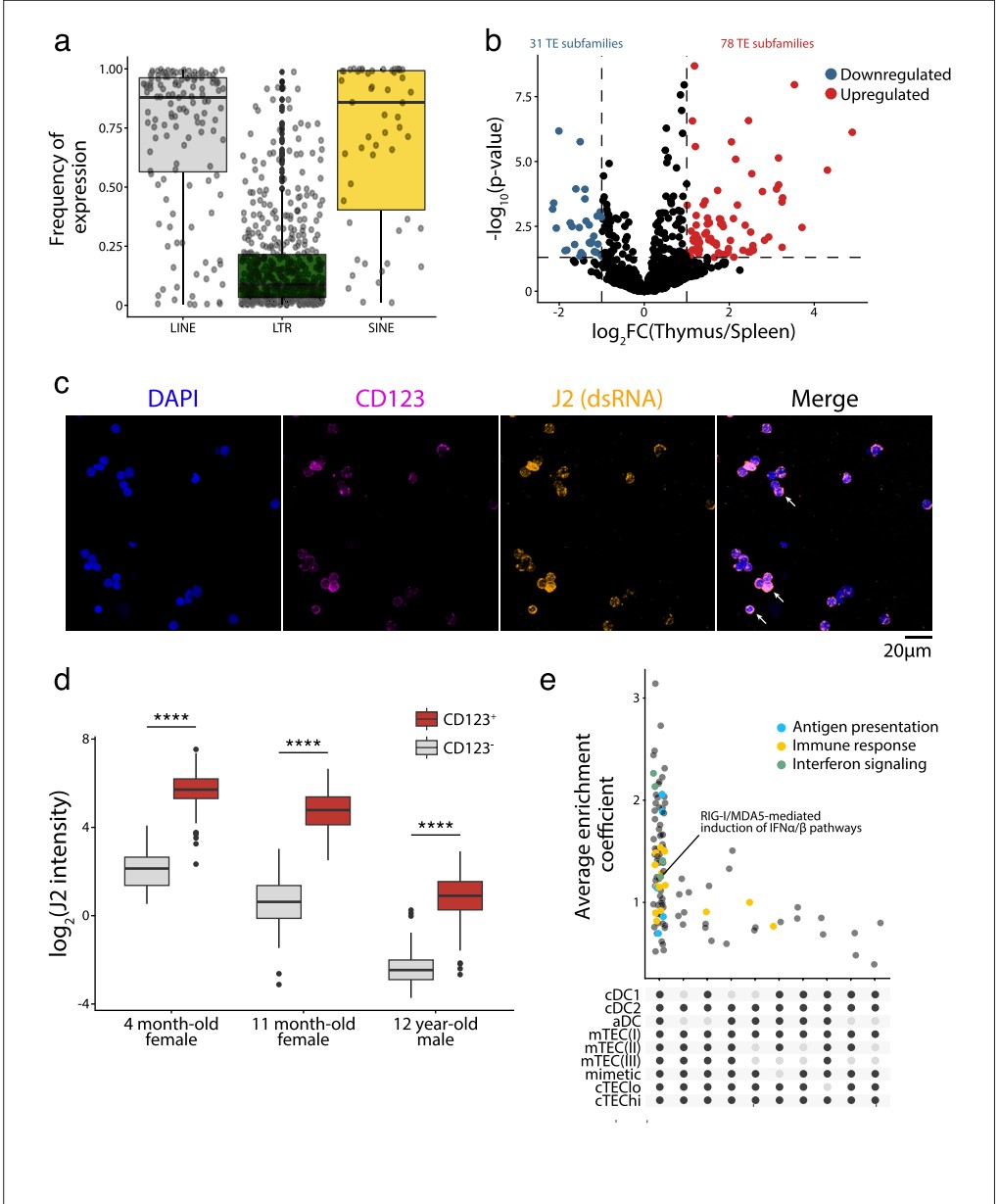

**Figure 4.** Transposable element (TE) expression in human plasmacytoid dendritic cells (pDCs) is associated with dsRNA formation and type I IFN signaling. (**a**) Frequency of long interspersed nuclear elements (LINE), long terminal repeat (LTR), and short interspersed nuclear elements (SINE) subfamilies expression in thymic pDCs. (**b**) Differential expression of TE subfamilies between splenic and thymic pDCs. TE subfamilies significantly upregulated or downregulated by thymic pDCs are indicated in red and blue, respectively (Upregulated, $\log_2$(Thymus/Spleen) ≥ –1 and adj. p≤0.05; Downregulated, $\log_2$(Thymus/Spleen) ≤ –1 and adj. p≤0.05). (**c, d**) Immunostaining of dsRNAs in human thymic pDCs (CD123$^+$) using the J2 antibody (n = 3). (**c**) One representative experiment. Three examples of CD123 and J2 colocalization are shown with white arrows. (**d**) J2 staining intensity in CD123$^+$ and CD123$^-$ cells from three human thymi (Wilcoxon rank-sum test, ****p-value≤0.0001). (**e**) UpSet plot showing gene sets enriched in pDCs compared to the other populations of thymic antigen-presenting cells (APCs). On the lower panel, black dots represent cell populations for which gene signatures are significantly depleted compared to pDCs. All comparisons where gene signatures were significantly enriched in pDCs are shown.

The online version of this article includes the following figure supplement(s) for figure 4:

**Figure supplement 1.** Transposable element (TE) expression in human splenic plasmacytoid dendritic cells (pDCs).

**Figure supplement 2.** A higher proportion of reads originates from transposable elements (TEs) in plasmacytoid dendritic cells (pDCs) than in other thymic antigen-presenting cells (APCs).

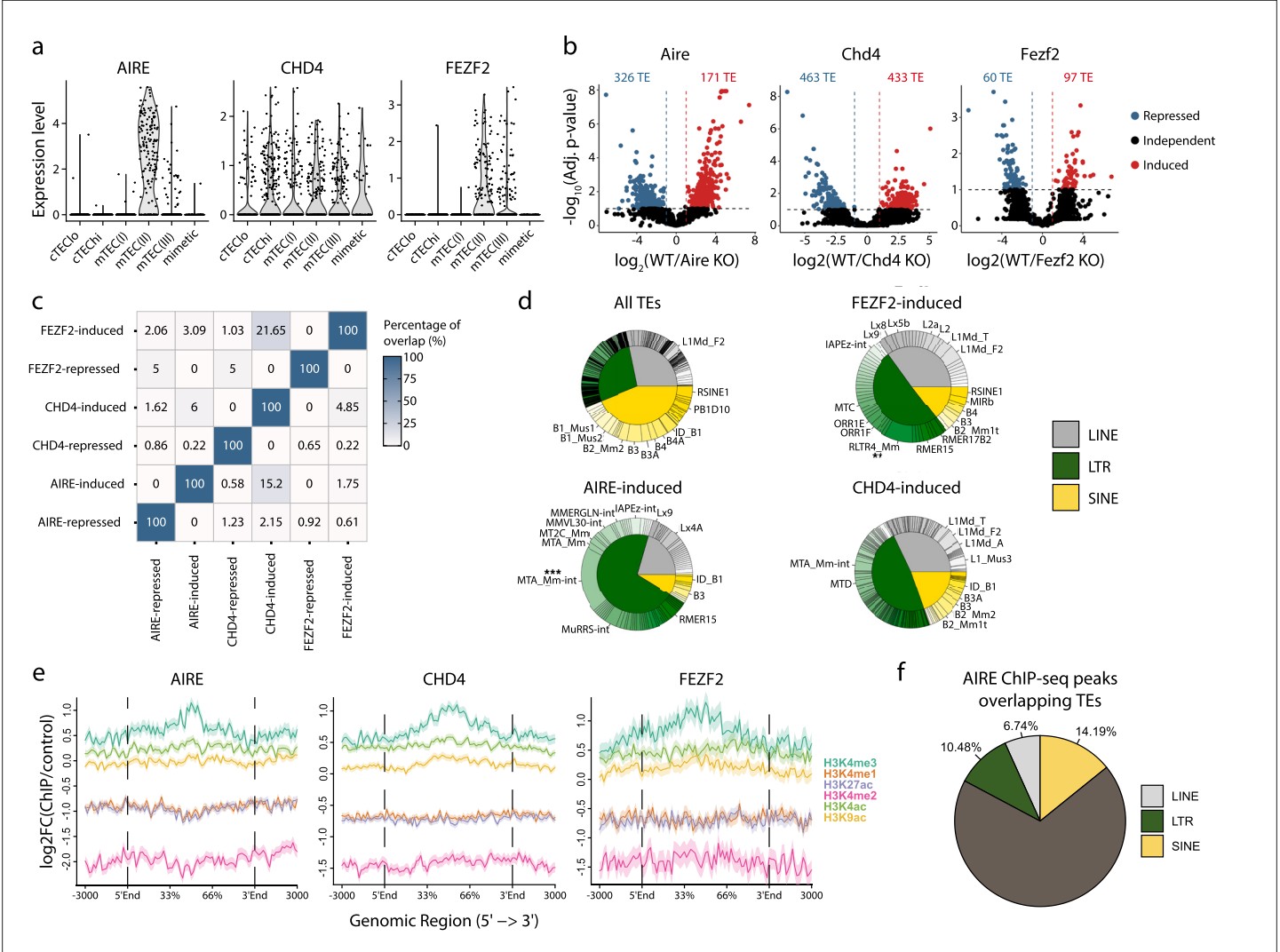

**Figure 5.** *AIRE*, *FEZF2*, and *CHD4* regulate non-redundant sets of transposable elements (TEs) in murine medullary thymic epithelial cells (mTECs). (**a**) Expression of *AIRE*, *CHD4*, and *FEZF2* in human TEC subsets. (**b**) Differential expression of TE loci between wild-type (WT) and *Aire-*, *Chd4-*, or *Fezf2-*knockout (KO) mice (Induced, $\log_2$(WT/KO) $\geq 2$ and adj. $p\leq0.05$; Repressed, $\log_2$(WT/KO) $\leq -2$ and adj. $p\leq0.05$). p-Values were corrected for multiple comparisons with the Benjamini–Hochberg procedure. The numbers of induced (red) and repressed (blue) TE loci are indicated on the volcano plots. (**c**) Overlap of TE loci repressed or induced by AIRE, FEZF2, and CHD4. (**d**) Proportion of TE classes and subfamilies in the TE loci regulated by AIRE, FEZF2, or CHD4, as well as all TE loci in the murine genome for comparison (chi-squared tests with Bonferroni correction, **adj. $p\leq0.01$, ***adj. $p\leq0.001$). (**e**) Plots for the tag density of H3K4me3 and H3K4me2 on the sequence and flanking regions (3000 base pairs) of TE loci induced by AIRE, FEZF2, and CHD4. (**f**) Proportion of statistically significant peaks overlapping TE sequences in AIRE ChIP-seq data from murine mTECs.

The online version of this article includes the following figure supplement(s) for figure 5:

**Figure supplement 1.** Characterization of transposable element (TE) subfamilies regulated by AIRE, CHD4, and FEZF2 in murine medullary thymic epithelial cells (mTECs).

suggests that AIRE, CHD4, and FEZF2 nonrandomly affect the expression of TE sequences located in specific genomic regions. We observed no significant differences in the genomic localization of TE loci targeted by AIRE, CHD4, and FEZF2 relative to the genomic localization of all TE sequences in the murine genome: most TE loci were located in intronic and intergenic regions (***Figure 5—figure supplement 1d***). Enrichment for intronic TEs could not be ascribed to induction of global intron retention: the intron retention ratio was similar for TEs regulated or not by AIRE, CHD4, and FEZF2 (***Figure 5—figure supplement 1e***). ChIP-seq-based analysis of permissive histone marks showed that TE loci induced by AIRE, CHD4, and FEZF2 were all marked by H3K4me3 (***Figure 5e***). As a proof of concept, we validated that 31.42% of AIRE peaks overlap with TE sequences by reanalyzing ChIP-seq

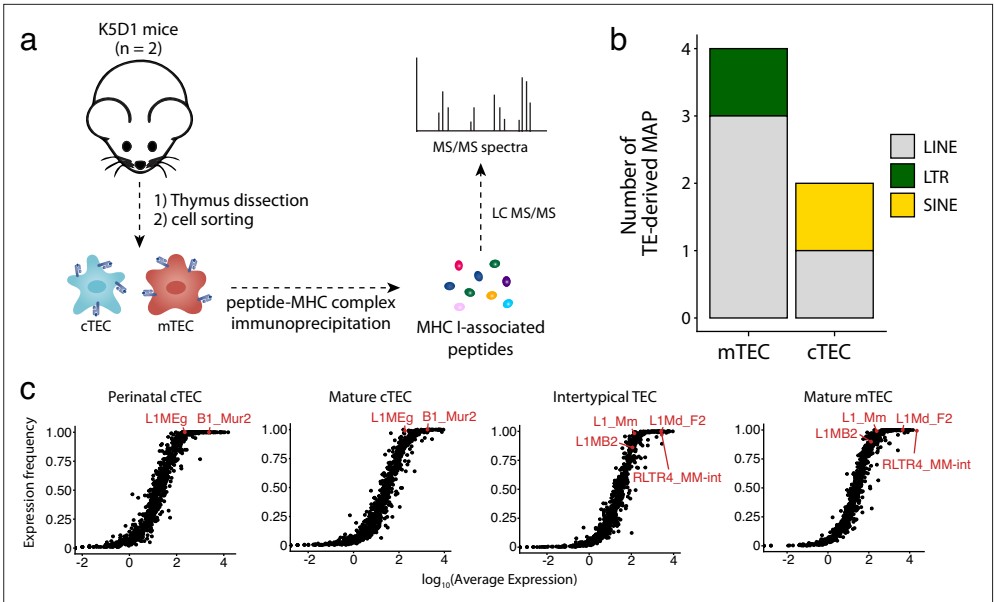

**Figure 6.** Murine cortical thymic epithelial cells (cTEC) and medullary thymic epithelial cell (mTEC) present transposable element (TE) MAPs. (**a**) mTECs and cTECs were isolated from the thymi of K5D1 mice (n = 2). The peptide-MHC I complexes were immunoprecipitated independently for both populations, and MAPs were sequenced by MS analyses. (**b**) Number of long interspersed nuclear elements (LINE)-, long terminal repeat (LTR)-, and short interspersed nuclear elements (SINE)-derived MAPs in mTECs and cTECs from K5D1 mice. (**c**) Distributions of TE subfamilies in murine TECs subsets based on expression level (x-axis) and frequency of expression (y-axis).

data, confirming AIRE's potential to bind TE sequences (*Figure 5f*). Hence, AIRE, CHD4, and FEZF2 regulate the expression of small yet non-redundant repertoires of TE sequences associated with permissive histone marks.

## TEs are translated and presented by MHC class I molecules in murine TECs

Several TEs are translated and generate MAPs (*Larouche et al., 2020*). Hence, the expression of TEs in cTECs and even more in mTECs raises a fundamental question: do these TEs generate MAPs that would shape the T cell repertoire? Mass spectrometry (MS) is the only method that can faithfully identify MAPs (*Shapiro and Bassani-Sternberg, 2023*; *Vizcaíno et al., 2020*; *Kubiniok et al., 2022*). Despite its quintessential role in central tolerance, the MAP repertoire of mTECs has never been studied by MS because of the impossibility of obtaining sufficient mTECs for MS analyses: mTECs represent ≤1% of thymic cells, and they do not proliferate in vitro. To get enough cTECs and mTECs for MS analyses, we used transgenic mice that express cyclin D1 under the control of the keratin 5 promoter (K5D1 mice). These mice develop dramatic thymic hyperplasia, but their thymus is morphologically and functionally normal (*Robles et al., 1996*; *Klug et al., 2000*; *Ohigashi et al., 2019*). Primary cTECs and mTECs (two replicates of 70 × 10⁶ cells from 121 and 90 mice, respectively) were isolated from the thymi of K5D1 mice as described (*Dumont-Lagacé et al., 2019*). Following cell lysis and MHC I immunoprecipitation, MAPs were analyzed by liquid chromatography MS/MS (*Figure 6a*). To identify TE-coded MAPs, we generated a TE proteome by in silico translation of TE transcripts expressed by mTECs or cTECs, and this TE proteome was concatenated with the canonical proteome. MS analyses enabled the identification of a total of 1636 and 1714 MAPs in mTECs and cTECs, respectively. From these, we identified four TE-derived MAPs in mTECs and two in cTECs, demonstrating that TEs can be translated and presented by MHC I in the thymic cortex and medulla (*Figure 6b* and *Supplementary file 1j*). These MAPs were coded by the three major groups of TE: LINEs (n = 1), LTRs (n = 1), and SINEs (n = 4). Next, we evaluated whether the low number of TE MAPs identified could result from mass spectrometry detection limits (*Ghosh et al., 2020*; *Nanaware et al., 2021*). We measured the level and frequency of TE expression in two subsets of cTECs (*Figure 6c*,

left) or mTECs (*Figure 6c*, right) using scRNA-seq data from *Baran-Gale et al., 2020*. TE subfamilies generating MAPs in cTECs or mTECs are highlighted in red in their respective plots. Strikingly, TECs highly and ubiquitously expressed the MAP-generating TE subfamilies. These results suggest that the contribution of TEs to the MAP repertoire of cTECs and mTECs might be significantly underestimated by the limits of detection of MS. This is particularly true for mTECs because they express high levels of TEs (*Figure 3d*), but their TE profile displays considerable intercellular heterogeneity (*Figure 3e*, *Figure 3—figure supplement 2*). Nonetheless, our data provide direct evidence that TEs can generate MAPs presented by cTECs and mTECs, which can contribute to thymocyte education.

## Discussion

TEs are germline-integrated parasitic DNA elements that comprise about half of mammalian genomes. Over evolutionary timescales, TE sequences have been co-opted for host regulatory functions. Mechanistically, TEs encode proteins and noncoding RNAs that regulate gene expression at multiple levels (*Bourque et al., 2018*; *Frank and Feschotte, 2017*). Regulation of IFN signaling and triggering innate sensors are the best-characterized roles of TEs in the mammalian immune system (*Kassiotis, 2023*). TEs are immunogenic and can elicit adaptive immune responses implicated in autoimmune diseases (*Larouche et al., 2020*; *Kassiotis, 2023*; *Gröger and Cynis, 2018*; *Volkman and Stetson, 2014*). Pervasive TE expression in various somatic organs means that co-evolution with their host must depend on establishing immune tolerance, a concept supported by the highly diversified TE repertoire expressed in mTECs (*Larouche et al., 2020*). This observation provided the impetus to perform multiomic studies of TE expression in the thymus. At the whole organ level, we found that TE expression showed extensive age- and cell lineage-related variations and was negatively correlated with cell proliferation and expression of KZFPs. The negative correlation between TE expression and cell cycle scores in the thymus is coherent with recent data showing that transcriptional activity of L1s is increased in senescent cells (*De Cecco et al., 2019*). A potential rationale for this could be to prevent deleterious transposition events during DNA replication and cell division. On the other hand, the contribution of KZFPs to TE regulation in the thymus is likely underestimated due to their typically low expression (*Huntley et al., 2006*) and scRNA-seq detection limit. Additionally, TEs interact with multiple TFs in all thymic cell subsets. This is particularly true for the LINE and SINE subfamilies that occupy larger genomic spaces. Notably, TEs appear to play particularly important roles in two cell types located in the thymic medulla: mTECs and pDCs.

As mTECs are the APC population crucial to central tolerance induction, their high and diverse TE expression is poised to impact the T cell repertoire's formation profoundly. The extent and complexity of TF-TE interactions were higher in mTECs than in all other thymic cell subsets. These interactions included *PAX1* and subunits of the NF-κB complex (e.g., *RELB*). PAX1 is essential for the development of TEC progenitors (*Yamazaki et al., 2020*), and *RELB* is for the development and differentiation of mTECs (*Mouri et al., 2014*). RelB-deficient mice have reduced thymic cellularity, markedly fewer mTECs, lack *Aire* expression, and suffer from autoimmunity (*Akiyama et al., 2008*; *O'Sullivan et al., 2018*). Under the influence of *Aire, Fezf2,* and *Chd4,* mTECs collectively express almost the entire exome (*Sansom et al., 2014*; *Pierre et al., 2017*). However, the expression of all genes in each mTEC would cause proteotoxic stress (*Pierre et al., 2017*). Hence, promiscuous expression of tissue-restricted genes in mTECs adopts a mosaic pattern: individual tissue-restricted genes are expressed in a small fraction of mTECs (*Michelson et al., 2022*; *Klein et al., 2014*). The present work shows that mTECs also express an extensive repertoire of TEs in a mosaic pattern (i.e., with considerable intercellular heterogeneity). Aire, Fezf2, and Chd4 regulate non-redundant sets of TEs and preferentially induce TE sequences associated with permissive histone marks. The immunopeptidome of thymic stromal cells is responsible for thymocyte education and represents one of the most fundamental 'known unknowns' in immunology. Inferences on the immunopeptidome of thymic stromal cells are based on transcriptomic data. However, (i) TCRs interact with MAPs, not transcripts, and (ii) the MAP repertoire cannot be inferred from the transcriptome (*Shapiro and Bassani-Sternberg, 2023*; *Caron et al., 2011*; *Admon, 2023*). Using K5D1 mice presenting prominent thymic hyperplasia, we conducted MS searches of TE MAPs, identifying four TE MAPs in mTECs and two in cTECs. These results demonstrate that cTECs and mTECs present TE MAPs and suggest they present different TE MAPs. However, the correlation between transcriptomic and immunopeptidomic data suggests that TECs can present many more TE MAPs. Their profiling will require MS analyses of enormous numbers

of TECs or the development of more sensitive MS techniques. As TE MAPs have been detected in normal and neoplastic extrathymic cells (*Larouche et al., 2020*; *Laumont et al., 2018*; *Burbage et al., 2023*; *Shah et al., 2023*), the presentation of TEs by mTECs is likely essential to central tolerance. In line with vibrant plaidoyers for a collaborative Human Immunopeptidome Project (*Vizcaíno et al., 2020*; *Shao et al., 2018*), our work suggests that immunopeptidomic studies should not be limited to protein-coding genes (2% of the genome) but also encompass non-coding sequences such as TEs.

The second population of cells exhibiting high TE expression, pDCs, are mainly seen as producers of IFN α/β and potentially as APCs (*Ginhoux et al., 2022*). Thymic and extrathymic pDCs are ontogenically and functionally different. They develop independently from each other from different precursor cells (*Lavaert et al., 2020*; *Le et al., 2020*; *Weijer et al., 2002*). IFN α/β secretion is inducible in extrathymic pDCs but constitutive in thymic pDCs (*Ginhoux et al., 2022*; *Colantonio et al., 2011*). In line with the location of pDCs in the thymic medulla, their constitutive IFN α/β secretion is instrumental in the terminal differentiation of thymocytes and the generation of Tregs and innate CD8 T cells (*Xing et al., 2016*; *Hanabuchi et al., 2010*; *Martín Gayo et al., 2010*; *Martinet et al., 2015*; *Epeldegui et al., 2015*). We report here that high TE expression is also a feature of thymic, but not extrathymic, pDCs. Thus, the present study provides a rationale for the constitutive IFN α/β secretion by thymic pDCs: they homogeneously express large numbers of TEs (in particular LINEs and SINEs), leading to the formation of dsRNAs that trigger RIG-I and MDA5 signaling that causes the constitutive secretion of IFN α/β. As such, our data suggest that recognition of TE-derived dsRNAs by innate immune receptors promotes a pro-inflammatory environment favorable to the establishment of central tolerance in the thymic medulla.

At first sight, the pleiotropic effects of TEs on thymic function may look surprising. It should be reminded that the integration of genetic parasites such as TEs is a source of genetic conflicts with the host. Notably, the emergence of adaptive immunity gave rise to higher-order conflicts between TEs and their vertebrate hosts (*Kassiotis, 2023*; *Boehm et al., 2023*). The crucial challenge for the immune system is developing immune tolerance towards TEs to prevent autoimmune diseases that affect up to 10% of humans (*Harroud and Hafler, 2023*) without allowing selfish retrotransposition events that hinder genome integrity. The resolution of these conflicts has been proposed to be a determining factor in shaping the function of the immune system (*Boehm et al., 2023*). Our data suggest that the thymus is the central battlefield for conflict resolution between TEs and T cells in vertebrates. Consistent with the implication of TEs in autoimmunity, more than 90% of putative causal variants associated with autoimmune diseases are in allegedly noncoding regions of the genome (*Harroud and Hafler, 2023*). In this context, our study illustrates the complexity of interactions between TEs and the vertebrate immune system and should provide impetus to explore them further in health and disease. We see two limitations to our study. First, as with all multiomic systems immunology studies, our work provides a roadmap for many future mechanistic studies that could not be realized at this stage. Second, our immunopeptidomic analyses of TECs prove that TECs present TE MAPs but certainly underestimate the diversity of TE MAPs presented by cTECs and mTECs.

## Materials and methods
### Experimental design
This study aimed to understand better the impacts of TE expression on thymus development and function. Thymic populations are complex and heterogeneous, so we opted for single-cell RNA-seq data to draw a comprehensive profile of TE expression in the thymus. To better understand the impact of AIRE, FEZF2, and CHD4 on TE expression in the mTEC(II) population, RNA-seq data from WT and KO murine mTEC, as well as ChIP-seq for different histone marks in murine mTECs, were reanalyzed to characterize the TE sequences regulated by these three proteins. Unless stated otherwise, studies were done in human cells. For MS analyses, two replicates of 70 million cells from K5D1 mice (*Robles et al., 1996*) were injected for both cTECs and mTECs. All experiments in mice were in accordance with the Canadian Council on Animal Care guidelines and approved by the *Comité de Déontologie de l'Expérimentation sur des Animaux* of Université de Montréal. Primary human thymi were obtained from 4-month-old to 12-year-old children undergoing cardiovascular surgery at the CHU Sainte-Justine. This project was approved by the CHU Sainte-Justine Research Ethics Board (protocol

and biobank #2126), which also granted permission to publish results. Written informed consent was obtained from parents of all children involved.

## Transcriptomic data processing

Preprocessing of the scRNA-seq data was performed with kallisto (*Bray et al., 2016*), which uses an expectation-maximization algorithm to reassign multimapping reads based on the frequency of unique mappers at each sequence and bustools workflow. For human thymic data from *Park et al., 2020* and splenic data from *Madissoon et al., 2019*, two different indexes were built for the pseudoalignment of reads with kallisto (version 0.46.0): one containing Ensembl 88 (GRCh38.88) transcripts used for the annotation of cell populations, and a second containing Ensembl 88 transcripts and human TE sequences (LINE, LTR, SINE) from RepeatMasker (*Smit et al., 2013*) which was used for all subsequent analyses of TE expression. For murine data from *Baran-Gale et al., 2020*, cell-type annotations from the original publication were used, and an index containing mm10 transcripts and murine TE sequences from RepeatMasker was used to analyze TE expression. The cell barcodes were corrected, and the feature-barcode matrices were generated with the correct count functions of bustools (version 0.39.3) (*Melsted et al., 2019*). For murine bulk RNA-seq data, an index composed of mm10 (GRCm38) transcripts and murine TE sequences from RepeatMasker was used for quantification with kallisto.

## ChIP-seq data reanalysis

ChIP-seq data for (i) ETS1 in human NK cells, (ii) AIRE in murine mTECs, and (iii) several histone marks of mTECs from WT mice were reanalyzed (see 'Data availability statement' for the complete list). ETS1 ChIP-seq reads were aligned to the reference *Homo sapiens* genome (GRCh38) using bowtie2 (version 2.3.5) (*Langmead and Salzberg, 2012*) with the `--very-sensitive` parameter. Multimapping reads were removed using the samtools view function with the -q 10 parameter, and duplicate reads were removed using the samtools markdup function with the -r parameter (*Danecek et al., 2021*). Peak calling was performed with macs2 with the -m 5 50 parameter (*Zhang et al., 2008*). Peaks overlapping with the ENCODE blacklist regions (*Amemiya et al., 2019*) were removed with bedtools intersect (*Quinlan and Hall, 2010*) with default parameters. Overlap of ETS1 peaks with TE sequences was determined using bedtools intersect with default parameters. BigWig files were generated using the bamCoverage function of deeptools2 (*Ramírez et al., 2016*), and genomic tracks were visualized in the USCS Genome Browser (*Kent et al., 2002*). For the murine histone marks and AIRE data, reads were aligned to the reference *Mus musculus* genome (mm10) using bowtie2 with the `--very-sensitive` parameter. Multimapping reads were removed using the samtools view function with the -q 10 parameter, and duplicate reads were removed using the samtools markdup function with the -r parameter. For histone marks, read coverage at the sequence body and flanking regions (±3000 base pairs) of TE loci induced by AIRE, FEZF2, and CHD4 was visualized using ngs.plot.r (version 2.63) (*Shen et al., 2014*). For AIRE, peaks overlapping with the ENCODE blacklist regions were removed with bedtools intersect, and overlap of peaks with TE sequences was determined using bedtools intersect with default parameters.

## Cell population annotation

Feature-barcode matrices were imported in R with SingleCellExperiment (version 1.12.0) (*Amezquita et al., 2020*). As a quality control, cells with less than 2000 UMI detected, less than 500 genes detected, or more than 5% reads assigned to mitochondrial genes were considered low quality and removed from the dataset with scuttle (version 1.0.4) (*McCarthy et al., 2017*). Cells with more than 7000 genes detected were considered doublets and removed. Normalization of cell size factors was performed with scran (version 1.18.7) (*Lun et al., 2016*), and log-normalization of read counts was done with scuttle with default parameters. Variable regions of TCR and IG genes, as well as ribosomal and cell cycle genes (based on *Park et al., 2020*), were removed, and highly variable features were selected based on a mean-variance trend based on a Poisson distribution of noise with scran. Adjustment of sequencing depths between batches and mutual nearest neighbors (MNN) correction were computed with batchelor (version 1.6.3) (*Haghverdi et al., 2018*). Cell clustering was performed with scran using the Jaccard index for edge weighting and the Louvain method for community detection. Lists of marker genes for human thymic cell populations and TEC subsets were taken from *Park et al.,*

*2020* and *Bautista et al., 2021*, whereas marker genes of splenic populations were based on *Madissoon et al., 2019*.

## TE expression throughout thymic development

The expression of TE subfamilies was obtained by summing the read counts of loci based on the RepeatMasker annotations. For each TE subfamily in each cell population, expression levels amongst developmental stages were normalized by dividing them with the maximal expression value. Next, the Euclidean distance between each TE subfamily in each cell population (based on their normalized expression across developmental stages) was computed, followed by unsupervised hierarchical clustering. The tree was then manually cut into three clusters, and enrichment of LINE, LTR, and SINE elements in these three clusters was determined using Fisher's exact tests. The cluster assigned to each TE subfamily in each cell population was visualized in a circos plot using the circlize package (version 0.4.14) (*Gu et al., 2014*) in R, and the percentage of each cell population found in embryonic or postnatal samples. Finally, we computed the frequency that each TE family was assigned to the three clusters, and the maximal value was kept. As a control, a random distribution of the expression of 809 TE subfamilies in 18 cell populations was generated. A cluster (cluster 1, 2, or 3) was randomly attributed for each combination of TE subfamily and cell type, and the maximal occurrence of a given cluster across cell types was then computed for each TE subfamily. Finally, the LINE, LTR, and SINE element distributions were compared to the random distribution with Kolmogorov–Smirnov tests.

## Regulation of TE expression by cell proliferation and KZFPs

Proliferation scores were generated for each dataset cell using the CellCycleScoring function of Seurat (version 4.1.0). As per *Cowan et al., 2019*, we combined previously published lists of G2M and S phase marker genes (*Kowalczyk et al., 2015*) to compute the proliferation scores. For each thymic cell population, we calculated the Spearman correlation between proliferation scores and the expression of TE subfamilies. The Benjamini–Hochberg method was applied to correct for multiple comparisons. Correlations were considered positive if the correlation coefficient was ≥0.2 and the adjusted p-value≤0.05, and negative if the coefficient was ≤–0.2 and the adjusted p-value≤0.05. We also computed the median of all correlation coefficients for each cell population. We then assigned the class of each TE subfamily correlated with cell proliferation and compared this distribution to the distribution of classes of all TE subfamilies in the human genome. The percentage of overlap of the sets of TE subfamilies significantly correlated with cell proliferation was determined. A list of 401 human KZFPs was downloaded from *Imbeault et al., 2017*. Spearman correlations between KZFP and TE expression were independently computed in each cell population with the same methodology as the cell proliferation analysis, and Benjamini–Hochberg correction for multiple comparisons was applied. The information on the enrichment of KZFPs within TE subfamilies was downloaded from *Imbeault et al., 2017*. Sharing of KZFP-TE pairs between cell populations was represented using the circlize package.

## Estimation of TE sequences' age

The sequence divergence (defined as the number of mismatches per thousand) was given by the milliDiv value in RepeatMasker. The milliDiv values of each TE locus were divided by the substitution rate of its host's genome ($2.2 \times 10^{-9}$ mutation/year for *Homo sapiens* and $4.5 \times 10^{-9}$ mutation/year for *Mus musculus*; *Lander et al., 2001*; *Waterston et al., 2002*). Finally, the age of each TE subfamily was determined by averaging the age of all loci of the subfamily.

## Interactions between TE subfamilies and transcription factors

We downloaded a list of 1638 transcription factors (TFs) manually curated by *Lambert et al., 2018*. For each cell population of the thymus, Spearman correlations were computed for each possible pair of TF and TE subfamily, and the Benjamini–Hochberg method was applied to correct the p-values for multiple comparisons. Correlations were considered significant if (i) the correlation coefficient was ≥0.2, (ii) the adjusted p-value was ≤0.05, and (iii) the TF was expressed by ≥10% of the cells of the population. The correlations were validated using a bootstrap procedure (1000 iterations) to ensure their reproducibility. Briefly, we randomly selected n cells out of the n cells of a given population (while allowing cells to be selected multiple times). The empirical p-value was determined by dividing

the number of iterations with a correlation coefficient <0.2 by the total number of iterations (1000). In parallel, the curated binding motifs of 945 TFs were downloaded from the JASPAR database. We then used the *Find Individual Motif Occurrences* (FIMO) software (*Grant et al., 2011*) to identify the 100,000 genomic positions with the most significant matches for the TF binding motif. These lists of binding motif positions were then intersected with the positions of TE loci with the intersect function of BEDTools (version 2.29.2) (*Quinlan and Hall, 2010*), and the percentage of TE loci of each subfamily harboring TF binding motifs was determined. Thus, in a specific cell population of the thymus, a TF was considered as interacting with a TE subfamily if it satisfied two criteria: (i) its expression was correlated with the one of the TE family (Spearman coefficient ≥ 0.2, adjusted p-value≤0.05, and expression of TF in ≥10% of cells), and (ii) at least one locus of the TE subfamily contained a binding motif of the TF. For each cell population, networks of interactions between TF and TE subfamilies were generated with the network package (version 1.17.1) (*Butts, 2008*) in R and represented with the ggnetwork package. For the sake of clarity, only the most significant interactions were illustrated for each cell type (i.e., correlation coefficient ≥ 0.3, TF binding sites in ≥1% of the loci of the TE subfamily, and TF expression in ≥10% of cells of the population). Sharing of TF-TE interactions between cell populations was represented with a chord diagram using the circlize package. For each TE subfamily, the number of interactions with TFs and the number of loci of the TE subfamily in the human genome were determined. Wilcoxon–Mann–Whitney tests were used to compare the number of interactions with TF of LTR, LINE, and SINE elements, whereas Kendall tau correlation was calculated between the number of interactions with TF and the number of loci of TE subfamilies.

## Identification of TE promoter and enhancer candidates

From the previously identified list of TF-TE interactions, we isolated the specific loci containing TF binding sites from the subfamilies whose expression was positively correlated with the TF. To determine if these TE loci could act as promoters or enhancers, we used histone ChIP-seq data from the ENCODE consortium for H3K27ac, H3K4me1, and H3K4me3. BED files from the ENCODE consortium were downloaded for eight immune cell populations: B cells, CD4 single positive T cells (CD4 SP), CD8 single positive T cells (CD8 SP), DCs, monocytes and macrophages (mono/macro), NK cells, Th17, and Treg. TE loci colocalizing with peaks in histone ChIP-seq data were identified using the intersect function of BEDTools (version 2.29.2). To be considered enhancer candidates, TE loci had to colocalize with H3K27ac and H3K4me1 but not H3K4me3. To be considered as promoter candidates, TE loci had to colocalize with H3K27ac and H3K4me3, but not H3K4me1, and be located at ≤1000 nucleotides from a transcription start site (TSS) annotated in the refTSS database (*Abugessaisa et al., 2019*).

## Diversity of TE expression

The human thymic scRNA-seq dataset was subsampled to retain only postnatal cells, as it was shown by *Taveirne et al., 2020* that thymic APCs are mainly found in postnatal samples. The diversity of TE sequences expressed by thymic populations was assessed using Shannon entropy. Using the vegan package (version 2.5–7) (*Tikhonov et al., 2020*) in R, two distinct Shannon entropy metrics were computed for each cell population. First, the Shannon entropy was computed based on the expression level (i.e., log(read count)) of TE subfamilies for each cell individually. The median entropy was calculated for each cell population. In parallel, the diversity of TE sequences expressed by an entire population was also assessed. For this purpose, a binary code was generated to represent the expression status of TE subfamilies in each cell (where 1 is expressed and 0 is not expressed). For each population separately, the binary codes of individual cells were summed to obtain the frequency of expression of each TE subfamily in the population, which was used to compute the Shannon entropy of TE sequences expressed by the population. A linear model was generated with the lm function of the stats package in R to summarize the data distribution. The deviation (Δy) from the observed population's TE diversity and the one expected by the linear model was computed for each cell population.

## TE expression in thymic APC

A differential expression analysis of TE subfamilies between the subsets of thymic APC was performed with the FindAllMarkers function with default parameters of Seurat (*Satija et al., 2015*) with the MAST model. Finally, the heterogeneity of TE expression inside thymic APC subsets was evaluated with the MetaCell package (version 0.3.5) (*Baran et al., 2019*). The composition of the metacells was validated

based on manual annotation (see the 'Single-cell RNA-seq preprocessing' section), and only metacells with >50% of cells belonging to the same subset of thymic APCs were kept. Differential expression of TE subfamilies between metacells was performed as described above, and the percentage of overlap between the sets of TEs overexpressed by the different metacells was computed.

## Isolation of human thymic pDCs and immunostaining of dsRNAs

Primary human thymi were obtained from 4-month-old to 12-year-old children undergoing cardio-vascular surgeries at the CHU Sainte-Justine. This project was approved by the CHU Sainte-Justine Research Ethics Board (protocol and biobank #2126). Thymi from 4-month-old to 12-year-old individuals were cryopreserved in liquid nitrogen in the following solution: 95% (PBS-5% Dextran 40 (Sigma-Aldrich)) – 5% DMSO (Fisher Scientific). Protocol for thymic pDCs isolation was based on *Stoeckle et al., 2013*. Briefly, thymic samples were cut in ~2 mm pieces, followed by three rounds of digestion (40 min, 180 RPM at 37°C) in RPMI 1640 (Gibco) supplemented with 2 mg/ml of Collagenase A (Roche) and 0.1 mg/ml of DNase I (Sigma-Aldrich). APCs were then enriched using Percoll (Sigma-Aldrich) density centrifugation (3500 × *g*, 35 min at 4°C), followed by an FBS cushion density gradient (5 ml of RPMI 1640 containing enriched APCs layered on 5 ml of heat-inactivated FBS [Invitrogen, 12483020], 1000 RPM for 10 min at 4°C) to remove cell debris. Finally, thymic pDCs were magnetically enriched using the QuadroMACS Separator (Miltenyi). Cells were stained with a CD303 (BDCA-2) MicroBead Kit (Miltenyi), and labeled cells were loaded on LS columns (Miltenyi) for magnetic-activated cell sorting.

Purified thymic pDCs were pipetted on poly-L-lysine (Sigma-Aldrich, 1:10 in dH$_2$O) coated 15μ-Slide 8 well (ibid) and incubated for 2 hr at 37°C in RPMI 1640 supplemented with 10% BSA (Sigma-Aldrich). Cells were fixed using 1% (w/v) paraformaldehyde (PFA, Sigma-Aldrich) in PBS 1× (Sigma-Aldrich) for 30 min at room temperature. Cells were permeabilized for 30 min at room temperature with 0.1% (v/v) Triton X-100 (Sigma-Aldrich) in PBS 1×, followed by blocking using 5% (w/v) BSA (Sigma-Aldrich) in PBS 1× for 30 min at room temperature. Immunostaining was performed in four steps to avoid unspecific binding of the secondary antibodies: (i) incubation overnight at 4°C with the mouse mono-clonal IgG2a J2 antibody anti-dsRNA (Jena Bioscience, Cat# RNT-SCI-10010500, dilution 1:200), (ii) incubation with the donkey anti-mouse IgG (H+L) antibody coupled to Alexa Fluor 555 (Invitrogen, Cat# A-31570, dilution 1:500) for 30 min at room temperature, (iii) incubation with the mouse mono-clonal IgG1 clone 6H6 anti-CD123 (eBioscience, Cat# 14-1239-82, 1:100) for 1 hr at room temperature, and (iv) incubation with the goat anti-mouse IgG1 polyclonal Alexa Fluor 488 antibody (Invitrogen, Cat# A-21121, 1:1000) for 30 min at room temperature. Finally, cells were stained with DAPI (Invitrogen, Cat# D3571, 1:1000) for 5 min at room temperature. All antibodies and DAPI were diluted in a blocking solution. Image acquisition was made with an LSM 700 laser scanning confocal microscope (Zeiss) using a ×40 oil objective (Zeiss, Plan-Neofluar N.A. 1.4) and the ZEN software. Using the white-TopHat function of the EBImage package and the sigmoNormalize function of the MorphoR package in R, the background of the DAPI signal was removed. The nuclei were segmented on the resulting images as circular shapes based on the DAPI signal. The mean intensity of CD123 and J2 staining was determined for each cytoplasm, defined as 19 nm rings around nuclei. Based on the distribution of the CD123 signal across cells, a threshold between CD123$^-$ and CD123$^+$ cells was set up for each replicate independently. J2 signal intensity was compared between CD123$^-$ and CD123$^+$ cells using the Wilcoxon rank-sum test in R.

## Gene set enrichment analysis

Gene set enrichment analyses were performed to determine which biological processes are enriched in mTEC(II) and pDCs. Differential gene expression analyses were performed between each possible pair of thymic APCs subsets using MAST with the FindMarkers function of Seurat. The gene set enrichment analysis was performed using the iDEA package (version 1.0.1) (*Ma et al., 2020*) in R. As per *Ma et al., 2020*, the fold change and standard error of gene expression were used as input for iDEA, in addition to predefined lists of gene sets compiled in the iDEA package. Gene sets associated with antigen presentation, interferon signaling, and immune response were manually annotated. iDEA was launched with default parameters, except for the 500 iterations of the Markov chain Monte Carlo algorithm, and p-values were corrected with the Louis method. We also visualized the expression of *AIRE, FEZF2,* and *CHD4* in the TEC lineage to validate their expression in mTEC(II).

## TE loci regulated by AIRE, FEZF2, and CHD4

A differential expression analysis of TE subfamilies between WT and *Aire*-, *Fezf2*-, or *Chd4*-KO mice was performed with the voom method of the limma package (version 3.46.0) (*Law et al., 2014*; *Ritchie et al., 2015*). Stringent criteria (i.e., an expression below 2 transcripts per million [TPM] in all samples) were applied to remove lowly expressed TEs. TE subfamilies with (i) a fold change ≥2 and an adjusted p-value≤0.05 or (ii) a fold change ≤–2 and an adjusted p-value≤0.05 were considered as induced and repressed, respectively. The percentage of overlap between the sets of TE loci induced or repressed by AIRE, FEZF2, and CHD4 was computed. The class and subfamily were assigned to each regulated TE locus, and the distributions of classes and subfamilies across all TE sequences of the murine genome were used as controls. Significant enrichment of classes or subfamilies was determined with chi-squared tests, and a Bonferroni correction for multiple comparisons was performed to enrich subfamilies in induced or repressed TEs. The distance between TE loci induced or repressed by AIRE, FEZF2, or CHD4 was defined as the minimal distance between the middle position of TE loci on the same chromosome. As a control, distributions of randomly selected TE loci whose expression is independent of AIRE, FEZF2, and CHD4 and equal size to the sets of regulated TEs were generated (e.g., if 433 TE loci are induced by CHD4, 433 independent TE loci were randomly selected). Wilcoxon rank-sum tests were used to compare random and regulated distributions. Genomic positions of exons, introns 3′ and 5′ untranslated transcribed region (UTR) were downloaded from the UCSC Table Browser. The genomic localization of regulated TEs was determined using the intersect mode of the BEDTools suite version 2.29.2. TE loci not located in exons, introns, 3′UTR, or 5′UTR were considered intergenic. The percentage of regulated TE loci in each type of genomic region was determined and compared to the genomic localization of all TE loci in the murine genome with chi-square tests. Finally, we estimated the frequency of intron retention events for introns containing TE loci regulated by AIRE, FEZF2, or CHD4 with S-IRFindeR (*Broseus and Ritchie, 2020*). Sequencing reads were aligned to the reference *Mus musculus* genome (mm10) using STAR version 2.7.1a (*Dobin et al., 2013*) with default parameters. Each intron's Stable Intron Retention ratio (SIRratio) was computed with the computeSIRratio function of S-IRFindeR. Introns containing TE loci induced by AIRE, FEZF2, or CHD4 were filtered using BEDTools intersect. Random distributions of equivalent sizes of introns containing TE sequences independent of AIRE, FEZF2, and CHD4 were generated as control. A SIRratio of 0.1 was used as a threshold of significant intron retention events.

## Enzymatic digestion and isolation of murine TECs

Thymic stromal cell enrichment was performed as previously described (*Dumont-Lagacé et al., 2019*; *Kim et al., 2015*). Briefly, thymi from 16- to 22-week-old K5D1 mice were mechanically disrupted and enzymatically digested with papain (Worthington Biochemical Corporation), DNase I (Sigma-Aldrich), and collagenase IV (Sigma-Aldrich) at 37°C. Next, the single-cell suspension obtained after enzymatic digestion was maintained at 4°C in FACS buffer (PBS, 0.5% [w/v] BSA, 2 mM EDTA) and enriched in thymic epithelial cells using anti-EpCAM (CD326) or anti-CD45 microbeads (mouse, Miltenyi) and LS columns (Miltenyi). Then, the enriched epithelial cell suspension was stained for flow cytometry cell sorting with the following antibodies and dyes: anti-EpCAM-APC-Cy7 clone G8.8 (BioLegend, Cat# 118218), anti-CD45-APC clone 30-F11 (BD Biosciences, Cat# 559864), anti-UEA1-biotinylated (Vector Laboratories, Cat# B-1065), anti-I-A/I-E-Alexa Fluor 700 clone M5/114.15.2 (BioLegend, Cat# 107622), anti-Ly51-FITC clone 6C3 (BioLegend, Cat# 553160), anti-streptavidin-PE-Cy7 (BD Biosciences, Cat# 557598), and 7-AAD (BD Biosciences, Cat# 559925). Cell sorting was performed using a BD FACSAria (BD Biosciences), and data were analyzed using the FACSDiva. TECs were defined as EpCAM$^+$CD45$^-$, while the cTEC and mTEC subsets were defined as UEA1$^-$Ly51$^+$ and UEA1$^+$Ly51$^-$ TEC, respectively.

## RNA-sequencing

Total RNA from 80,000 mTECs or cTECs was isolated using TRIzol and purified with an RNeasy micro kit (QIAGEN). Total RNA was quantified using Qubit (Thermo Scientific), and RNA quality was assessed with the Agilent 2100 Bioanalyzer (Agilent Technologies). Transcriptome libraries were generated using a KAPA RNA HyperPrep kit (Roche) using a poly(A) selection (Thermo Scientific). Sequencing was performed on the Illumina NextSeq 500, obtaining ~200 million paired-end reads per sample.

## Preparation of CNBR-activated Sepharose beads for MHC I immunoprecipitation

CNBR-activated Sepharose 4B beads (Sigma-Aldrich, Cat# 17-0430-01) were incubated with 1 mM HCl at a ratio of 40 mg of beads per 13.5 ml of 1 mM HCl for 30 min with tumbling at room temperature. Beads were spun at 215 × $g$ for 1 min at 4°C, and supernatants were discarded. 40 mg of beads were resuspended with 4 ml of coupling buffer (0.1 M NaHC0$_3$/0.5 M NaCl pH 8.3), spun at 215 × $g$ for 1 min at 4°C, and the supernatants were discarded. Mouse antibodies Pan-H2 (clone M1/42), H2-K$^b$ (clone Y-3), and H2-D$^b$ (clone 28-14-8S) were coupled to beads at a ratio of 1 mg of antibody to 40 mg of beads in coupling buffer for 120 min with tumbling at room temperature. Beads were spun at 215 × $g$ for 1 min at 4°C, and supernatants were discarded. 40 mg of beads were resuspended with 1 ml of blocking buffer (0.2 M glycine), incubated for 30 min with tumbling at room temperature, and the supernatants were discarded. Beads were washed by centrifugation twice with PBS pH 7.2, resuspended at a concentration of 1 mg of antibody per ml of PBS pH 7.2, and stored at 4°C.

## Immuno-isolation of MAPs

Frozen pellets of mTECs (90 mice, 191 million cells total) and cTECs (121 mice, 164 million cells total) were thawed, pooled, and resuspended with PBS pH 7.2 up to 4 ml and then solubilized by adding 4 ml of detergent buffer containing PBS pH 7.2, 1% (w/v) CHAPS (Sigma, Cat# C9426-5G) supplemented with Protease inhibitor cocktail (Sigma, Cat# P8340-5mL). Solubilized cells were incubated for 60 min with tumbling at 4°C and then spun at 16,600 × $g$ for 20 min at 4°C. Supernatants were transferred into new tubes containing 1.5 mg of Pan-H2, 0.5 mg of H2-K$^b$, and 0.5 mg of H2-D$^b$ antibodies covalently-cross-linked CNBR-Sepharose beads per sample and incubated with tumbling for 180 min at 4°C. Samples were transferred into Bio-Rad Poly prep chromatography columns and eluted by gravity. Beads were first washed with 11.5 ml PBS, then with 11.5 ml of 0.1× PBS, and finally with 11.5 ml of water. MHC I complexes were eluted from the beads by acidic treatment using 1% trifluoroacetic acid (TFA). Acidic filtrates containing peptides were separated from MHC I subunits (HLA molecules and β–2 microglobulin) using home-made stage tips packed with two 1 mm diameter octadecyl (C-18) solid phase extraction disks (EMPORE). Stage tips were pre-washed with methanol, then with 80% acetonitrile (ACN) in 0.1% TFA, and finally with 1% TFA. Samples were loaded onto the stage tips and washed with 1% TFA and 0.1% TFA. Peptides were eluted with 30% ACN in 0.1% TFA, dried using vacuum centrifugation, and then stored at –20°C until MS analysis.

## MS analyses

Peptides were loaded and separated on a home-made reversed-phase column (150 µm i.d. by 200 mm) with a 106 min gradient from 10 to 38% B (A: formic acid 0.1%; B: 80% CAN 0.1% formic acid) and a 600 nl/min flow rate on an Easy nLC-1200 connected to an Orbitrap Exploris 480 (Thermo Fisher Scientific). Each full MS spectrum acquired at a resolution of 240,000 was followed by tandem-MS (MS-MS) spectra acquisition on the most abundant multiply charged precursor ions for a maximum of 3 s. Tandem-MS experiments were performed using higher energy collision-induced dissociation (HCD) at a collision energy of 34%. The generation of the personalized proteome containing TE sequences, as well as the identification of TE-derived MAPs, was performed as per *Larouche et al., 2020* with the following modifications: the mm10 murine reference genome was downloaded from the UCSC Genome Browser, the annotations for murine genes and TE sequences were downloaded from the UCSC Table Browser, and the UniProt mouse database (16,977 entries) was used for the canonical proteome. MAPs were identified using PEAKS X Pro (Bioinformatics Solutions, Waterloo, ON). The level and frequency of expression of TE subfamilies generating MAPs or not were determined in thymic epithelial cells were determined by averaging the expression values across cells of a TEC subset and dividing the number of cells with a positive (i.e., >0) expression of the TEs by the total number of cells of the TEC subset, respectively.

## Acknowledgements

We thank Christian Charbonneau and Raphaëlle Lambert from IRIC's bio-imaging and genomics platforms, respectively. We also thank Allan Sauvat for the help with the microscopy quantification. We thank Mathilde Soulez, Bernhard Lehnertz, Biljana Culjkovic, Brian Wilhelm, and Michaël Imbeault

for insightful discussions. We are indebted to Kathie Béland and Elie Haddad from the CHU Sainte-Justine Research Center for providing the primary thymic samples.

## Additional information

### Funding

| Funder | Grant reference number | Author |
|---|---|---|
| Canadian Institutes of Health Research | FDN 148400 | Claude Perreault |

The funders had no role in study design, data collection and interpretation, or the decision to submit the work for publication.

### Author contributions

Jean-David Larouche, Conceptualization, Investigation, Visualization, Methodology, Writing - original draft, Project administration, Writing – review and editing; Céline M Laumont, Assya Trofimov, Conceptualization, Visualization, Methodology, Writing – review and editing; Krystel Vincent, Conceptualization, Methodology, Writing - original draft, Project administration, Writing – review and editing; Leslie Hesnard, Sylvie Brochu, Juliette F Humeau, Éric Bonneil, Joel Lanoix, Chantal Durette, Patrick Gendron, Jean-Philippe Laverdure, Investigation, Writing – review and editing; Caroline Côté, Investigation; Ellen R Richie, Resources, Writing – review and editing; Sébastien Lemieux, Pierre Thibault, Supervision, Writing – review and editing; Claude Perreault, Conceptualization, Supervision, Funding acquisition, Writing - original draft, Project administration, Writing – review and editing

### Author ORCIDs

Jean-David Larouche ⬡ http://orcid.org/0000-0003-3056-0714
Assya Trofimov ⬡ http://orcid.org/0000-0001-8748-3735
Jean-Philippe Laverdure ⬡ http://orcid.org/0000-0002-9245-6413
Claude Perreault ⬡ http://orcid.org/0000-0001-9453-7383

### Ethics

This project was approved by the CHU Sainte-Justine Research Ethics Board (protocol and biobank #2126). Written informed consent was obtained from parents of all 391 children involved.
This project was approved by the Comité de Déontologie de l'Expérimentation sur des Animaux of Université de Montréal (certificate 19-107).

Reviewer #1 (Public Review): https://doi.org/10.7554/eLife.91037.3.sa1
Reviewer #2 (Public Review): https://doi.org/10.7554/eLife.91037.3.sa2
Author response https://doi.org/10.7554/eLife.91037.3.sa3

## Additional files

### Supplementary files

• Supplementary file 1. Numerical data for TE expression in thymic development and function. (a) Expression of TE subfamilies throughout thymus development. (b) Correlation between TE expression and cell proliferation scores. (c) Correlation between KZFP and TE subfamily expression. (d) Interactions between TE and transcription factors. (e) ENCODE dataset – samples characteristics. (f) Characteristics of TE promoter and enhancer candidates. (g) Differential expression of TE between APC subsets. (h) Gene signatures enriched in pDC relative to other APC subsets. (i) Murine TE loci regulated by Aire, Chd4, and Fezf2. (j) Characteristics of the identified TE-derived MAP.

• MDAR checklist

### Data availability

scRNA-seq data of human thymi and spleen were downloaded from ArrayExpress (accession number E-MTAB-8581) and the NCBI BIOPROJECT (accession code PRJEB31843), respectively. scRNA-seq

data of murine thymi were downloaded from ArrayExpress (accession number E-MTAB-8560). RNA-seq data from WT, Aire-KO, Fezf2-KO, and Chd4-KO murine mTECs were downloaded from the Gene Expression Omnibus (GEO) under the accession code GSE144880. ChIP-seq data of ETS1 in human NK cells and AIRE in murine mTECs were downloaded from GEO (accession codes GSE124104 and GSE92654, respectively). ChIP-seq data for different histone marks in murine mTECs were also downloaded from GEO: H3K4me3 for mTECs (GSE53111); H3K4me1 and H3K27ac from MHCIIhi mTECs (GSE92597); H3K4me2 in mTEC-II (GSE103969); and H3K4ac and H3K9ac in mTECs (GSE114713). Transcriptomic and immunopeptidomic data of K5D1 mice mTECs and cTECs generated in this study are available on the Gene Expression Omnibus (GEO) under the accession GSE232011 and on the Proteomics Identification Database (PRIDE) under the accession PXD042241, respectively.

The following datasets were generated:

| Author(s) | Year | Dataset title | Dataset URL | Database and Identifier |
| --- | --- | --- | --- | --- |
| Courcelles M, Thibault P | 2024 | LC-MSMS of the MHC-I immunopeptidome of murine thymic epithelial cells | https://www.ebi.ac.uk/pride/archive/projects/PXD042241 | PRIDE, PXD042241 |
| Larouche JD, Perreault C | 2024 | Transposable elements regulate T cell maturation and education in the thymus | https://www.ncbi.nlm.nih.gov/geo/query/acc.cgi?acc=GSE232011 | NCBI Gene Expression Omnibus, GSE232011 |

The following previously published datasets were used:

| Author(s) | Year | Dataset title | Dataset URL | Database and Identifier |
| --- | --- | --- | --- | --- |
| Park JE | 2020 | A cell atlas of human thymic development defines T cell repertoire formation | https://www.ebi.ac.uk/biostudies/arrayexpress/studies/E-MTAB-8581?query=E-MTAB-8581 | EMBL-EBI ArrayExpress, E-MTAB-8581 |
| Madissoon E, Meyer KB | 2019 | Ischaemic sensitivity of human tissue by single cell RNA seq | https://www.ncbi.nlm.nih.gov/bioproject/?term=PRJEB31843 | NCBI BioProject, PRJEB31843 |
| Baran-Gale J, Morgan M | 2020 | Single-cell RNA-sequencing of mouse thymic epithelial cells across the first year of life | https://www.ebi.ac.uk/biostudies/arrayexpress/studies/E-MTAB-8560?query=E-MTAB-8560 | EMBL-EBI ArrayExpress, E-MTAB-8560 |
| Tomofuji Y, Takaba H, Suzuki HI, Benlaribi R, Peña Martinez CD, Abe Y, Morishita Y, Okamura T, Taguchi A, Kodama T, Takayanagi H | 2020 | RNA-seq, ChIP-seq, and ATAC-seq analyses of mTECs | https://www.ncbi.nlm.nih.gov/geo/query/acc.cgi?acc=GSE144880 | NCBI Gene Expression Omnibus, GSE144880 |
| Taveirne S, Wahlen S, Van Loocke W, Kiekens L, Persyn E, Van Ammel E, De Mulder K, Roels J, Tilleman L, Aumercier M, Matthys P, Van Nieuwerburgh F, Kerre TCC, Taghon T, Van Vlierberghe P, Vandekerckhove P, Leclercq G | 2020 | The transcription factor ETS1 is a master regulator of human NK cell differentiation | https://www.ncbi.nlm.nih.gov/geo/query/acc.cgi?acc=GSE124104 | NCBI Gene Expression Omnibus, GSE124104 |

*Continued on next page*

*Continued*

| Author(s) | Year | Dataset title | Dataset URL | Database and Identifier |
|-----------|------|---------------|-------------|-------------------------|
| Bansal K, Yoshida H, Benoist C, Mathis D | 2017 | Aire, guardian of immunological tolerance, binds to and activates super-enhancers | https://www.ncbi.nlm.nih.gov/geo/query/acc.cgi?acc=GSE92654 | NCBI Gene Expression Omnibus, GSE92654 |
| Sansom SN, Shikama-Dorn N, Zhanybekova S, Nusspaumer G, Heger A, Ponting CP, Hollander GA | 2014 | Aire secures the transcription of polycomb marked genes to complete a comprehensive program of promiscuous gene expression in the thymic epithelial cell lineage | https://www.ncbi.nlm.nih.gov/geo/query/acc.cgi?acc=GSE53111 | NCBI Gene Expression Omnibus, GSE53111 |
| Bansal K, Mathis D, Benoist C | 2017 | Aire, guardian of immunological tolerance, binds to and activates super-enhancers [ChIP-seq] | https://www.ncbi.nlm.nih.gov/geo/query/acc.cgi?acc=GSE92597 | NCBI Gene Expression Omnibus, GSE92597 |
| Handel AE, Holländer GA | 2018 | Comprehensively profiling the chromatin architecture of tissue restricted antigen expression in thymic epithelial cells | https://www.ncbi.nlm.nih.gov/geo/query/acc.cgi?acc=GSE114713 | NCBI Gene Expression Omnibus, GSE114713 |
| Amit I, Abramson J, Bornstein C, Nevo S, Giladi A, Kadouri N | 2018 | Large-scale single cell mapping of the thymic stroma identifies a new thymic epithelial cell lineage [ChIP-seq] | https://www.ncbi.nlm.nih.gov/geo/query/acc.cgi?acc=GSE103969 | NCBI Gene Expression Omnibus, GSE103969 |

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
