## [Editor Report · eLife assessment]

This **important** study shows, based on analyses of single-cell RNA-seq data sets of thymus cells, that transposable elements (TEs) are broadly expressed in thymic stromal cells, especially in medullary thymic epithelial cells and plasamacytoid dendritic cells. The authors also show that at least some TE-derived peptides are presented by MHC-I molecules in the thymus. The study provides **solid** findings supporting a role of TEs in thymic T-cell selection and immune self-tolerance.

---

## [Referee Report · Reviewer #1 (Public Review)]

Summary:

Transposable Elements (TEs) are exogenously acquired DNA regions that have played important roles in the evolutional acquisition of various biological functions. TEs may have been important in the evolution of the immune system, but their role in thymocytes has not been fully clarified.

Using the human thymus scRNA dataset, the authors suggest the existence of cell type-specific TE functions in the thymus. In particular, it is interesting to show that there is a unique pattern in the type and expression level of TEs in thymic antigen-presenting cells, such as mTECs and pDCs, and that they are associated with transcription factor activities. Furthermore, the authors suggested that TEs may be non-redundantly regulated in expression by Aire, Fezf2, and Chd4, and that some TE-derived products are translated and present as proteins in thymic antigen-presenting cells. These findings provide important insights into the evolution of the acquired immune system and the process by which the thymus acquires its function as a primary lymphoid tissue.

Strengths:

(1) By performing single-cell level analysis using scRNA-seq datasets, the authors extracted essential information on heterogeneity within the cell population. It is noteworthy that this revealed the diversity of expression not only of known autoantigens but also of TEs in thymic antigen-presenting cells.

(2) The attempt to use mass spectrometry to confirm the existence of TE-derived peptides is worthwhile, even if the authors did not obtain data on as many transcripts as expected.

(3) The use of public data sets and the clearly stated methods of analysis improved the transparency of the results.

Weaknesses:

(1) The authors sometimes made overstatements largely due to the lack or shortage of experimental evidence.

For example in Figure 4, the authors concluded that thymic pDCs produced higher copies of TE-derived RNAs to support the constitutive expression of type-I interferons in thymic pDCs, unlike peripheral pDCs. However, the data was showing only the correlation between the distinct TE expression pattern in pDCs and the abundance of dsRNAs. We are compelled to say that the evidence is totally too weak to mention the function of TEs in the production of interferon. Even if pDCs express a distinct type and amount of TE-derived transcripts, it may be a negligible amount compared to the total cellular RNAs. How many TE-derived RNAs potentially form the dsRNAs? Are they over-expressed in pDCs?

The data interpretation requires more caution to connect the distinct results of transcriptome data to the biological significance.

We contend that our manuscript combines the attributes of a research article (novel concepts) and a resource article (datasets of TEs implicated in various aspects of thymus function). The critical strength of our work is that it opens entirely novel research perspectives. We are unaware of previous studies on the role of TEs in the human thymus. The drawback is that, as with all novel multi-omic systems biology studies, our work provides a roadmap for a multitude of future mechanistic studies that could not be realized at this stage. Indeed, we performed wet lab experiments to validate some but not all conclusions: (i) presentation of TE-derived MAPs by TECs and (ii) formation of dsRNAs in thymic pDCs. In response to Reviewer #1, we performed supplementary analyses to increase the robustness of our conclusions. Also, we indicated when conclusions relied strictly on correlative evidence and clarified the hypotheses drawn from our observations. Regarding the Reviewer's questions about TE-derived dsRNAs, LINE, LTR, and SINE elements all have the potential to generate dsRNAs, given their highly repetitive nature and bi-directional transcription (1). As ~32% of TE subfamilies are overexpressed in pDCs, we hypothesized that these TE sequences might form dsRNA structures in these cells. To address the Reviewer's concerns regarding the amount of TE-derived RNAs among total cellular RNAs, we also computed the percentage of reads assigned to TEs in the different subsets of thymic APCs (see Reviewer 1 comment #4).

------

I appreciate the authors' efforts to improve the quality of this valuable paper. The additional data proposed by the authors enhanced the possibility that the non-negligible amount of RNAs in pDCs is derived from TE elements. Their biological roles and significance will be demonstrated in future research.

(2) Lack of generality of specific examples. This manuscript discusses the whole genomic picture of TE expression. In addition, one good way is to focus on the specific example to clearly discuss the biological significance of the acquisition of TEs for the thymic APC functions and the thymic selection.

In Figure 2, the authors focused on ETS-1 and its potential target genes ZNF26 and MTMR3, however, the significance of these genes in NK cell function or development is unclear. The authors should examine and discuss whether the distinct features of TEs can be found among the genomic loci that link to the fundamental function of the thymus, e.g., antigen processing/presentation.

We thank the Reviewer for this highly relevant comment. We investigated the genomic loci associated with NK cell biology to determine if ETS1 peaks would overlap with TE sequences in protein-coding genes' promoter region. Figure 2h illustrates two examples of ETS1 significant peaks overlapping TE sequences upstream of PRF1 and KLRD1. PRF1 is a protein implicated in NK cell cytotoxicity, whereas KLRD1 (CD94) dimerizes with NKG2 and regulates NK cell activation via interaction with the nonclassical MHC-I molecule HLA-E (2, 3). Thus, we modified the section of the manuscript addressing these results to include these new analyses: "Finally, we analyzed publicly available ChIP-seq data of ETS1, an important TF for NK cell development (4), to confirm its ability to bind TE sequences. Indeed, 19% of ETS1 peaks overlap with TE sequences (Figure 2g). Notably, ETS1 peaks overlapped with TE sequences (Figure 2h, in red) in the promoter regions of PRF1 and KLRD1, two genes important for NK cells' effector functions (2, 3)."

------

I am convinced by the authors' explanation that TE elements may contribute to the functions of NK cells.

However, since I have understood that the main topic of this paper is about the thymus and thymic antigen-presenting cells, the mention of NK cells seems abrupt and unconnected to me. NK cells are a type of innate lymphocyte that arise in the bone marrow, and thymus is dispensable for their development and function. The readers might expect to find something more fundamental regarding the function of the thymus and immunological tolerance.

(3) Since the deep analysis of the dataset yielded many intriguing suggestions, why not add a discussion of the biological reasons and significance? For example, in Figure 1, why is TE expression negatively correlated with proliferation? cTEC-TE is mostly postnatal, while mTEC-TE is more embryonic. What does this mean?

We thank the Reviewer for this comment. To our knowledge, the relationship between cell division and transcriptional activity of TEs has not been extensively studied in the literature. However, a recent study has shown that L1 expression is induced in senescent cells. We therefore added the following sentences to our Discussion: "The negative correlation between TE expression and cell cycle scores in the thymus is coherent with recent data showing that transcriptional activity of L1s is increased in senescent cells (5). A potential rationale for this could be to prevent deleterious transposition events during DNA replication and cell division." We also added several discussion points regarding the regulation of TEs by KZFPs to answer concerns raised by Reviewer 2 (see Reviewer 2 comment #1).

------

I agree on the possibility suggested by the authors.

(4) To consolidate the experimental evidence about pDCs and TE-derived dsRNAs, one option is to show the amount of TE-derived RNA copies among total RNAs. The immunohistochemistry analysis in Figure 4 requires additional data to demonstrate that overlapped staining was not caused by technical biases (e.g. uneven fixation may cause the non-specifically stained regions/cells). To show this, authors should have confirmed not only the positive stainings but also the negative staining (e.g. CD3, etc.). Another possible staining control was showing that non-pDC (CD303- cell fractions in this case) cells were less stained by the ds-RNA probe.

We thank the Reviewer for this suggestion. We computed the proportion of reads in each cell assigned to two groups of sequences known to generate dsRNAs: TEs and mitochondrial genes (1). These analyses showed that the proportion of reads assigned to TEs is higher in pDCs than other thymic APCs by several orders of magnitude (~20% of all reads). In contrast, reads derived from mitochondrial genes had a lower abundance in pDCs. We included these results in Figure 4 - figure supplement 2 and included the following text in the Results section "To evaluate if these dsRNAs arise from TE sequences, we analyzed in thymic APC subsets the proportion of the transcriptome assigned to two groups of genomic sequences known as important sources of dsRNAs, TEs and mitochondrial genes (1). Strikingly, whereas the percentage of reads from mitochondrial genes was typically lower in pDCs than in other thymic APCs, the proportion of the transcriptome originating from TEs was higher in pDCs (~22%) by several orders of magnitude (Figure 4 - figure supplement 2)." As a negative control for the immunofluorescence experiments, we used CD123- cells. Indeed, flow cytometry analysis of the magnetically enriched CD303+ fraction was around 90% pure, as revealed by double staining with CD123 and CD304 (two additional markers of pDCs): CD123- cells were also CD304-/lo, showing that these cells are non- pDCs. Thus, we decided to compare the dsRNA signal between CD123+ cells (pDCs) and CD123- cells (non-pDCs). The difference between CD123+ and CD123- cells was striking (Figure 4d).

------

Although the technical concerns about immunostaining were not resolved, it is understandable that it would be difficult to rerun the experiment since the authors used the precious human thymi as the experimental material. Immunostaining co-staining requires careful interpretation so that careful experimental setup is needed.

---

## [Referee Report · Reviewer #2 (Public Review)]

Summary:

Larouche et al show that TEs are broadly expressed in thymic cells, especially in mTECs and pDCs. Their data suggest a possible involvement of TEs in thymic gene regulation and IFN-alpha secretion. They also show that at least some TE-derived peptides are presented by MHC-I in the thymus.

Strengths:

The idea of high/broad TE expression in the thymus as a mechanism for preventing TE-mediated autoimmunity is certainly an attractive one, as is their involvement in IFN-alpha secretion therein. The analyses and experiments presented here are therefore a very useful primer for more in-depth experiments, as the authors point out towards the end of the discussion.

Weaknesses:

There are many dangers about analysing RNA-seq data at the subfamily level. Outputs may be greatly confounded by pervasive transcription, DNA contamination, and overlap of TEs with highly expressed genes. Whether TE transcripts are independent units or part of a gene also has important implications for the conclusions drawn. The authors have tried to mitigate against some of these issues, but they have not been completely ruled out.

---

## [Author Response]

The following is the authors’ response to the original reviews.

We greatly appreciate the reviewers' and editors' comments and suggestions on our manuscript "Transposable elements regulate thymus development and function." We performed additional analyses to validate our results and rephrased some manuscript sections according to the comments. We believe these changes significantly increase the solidity of our conclusions. Our point-by-point answer to the reviewers' and editors' comments is detailed below. New data and analyses are shown in Figure 1d, Figure 2g and h, Figure 5e and f, Figure 1 – figure supplement 1, Figure 2 – figure supplement 2, Figure 3 – figure supplement 1 and 2, Figure 4 – figure supplement 2, Figure 5 – figure supplement 1, as well as the corresponding text sections.

**Reviewer #1:**
(1) The authors sometimes made overstatements largely due to the lack or shortage of experimental evidence.For example in figure 4, the authors concluded that thymic pDCs produced higher copies of TE-derived RNAs to support the constitutive expression of type-I interferons in thymic pDCs, unlike peripheral pDCs. However, the data was showing only the correlation between the distinct TE expression pattern in pDCs and the abundance of dsRNAs. We are compelled to say that the evidence is totally too weak to mention the function of TEs in the production of interferon. Even if pDCs express a distinct type and amount of TE-derived transcripts, it may be a negligible amount compared to the total cellular RNAs. How many TE-derived RNAs potentially form the dsRNAs? Are they over-expressed in pDCs?The data interpretation requires more caution to connect the distinct results of transcriptome data to the biological significance.

We contend that our manuscript combines the attributes of a research article (novel concepts) and a resource article (datasets of TEs implicated in various aspects of thymus function). The critical strength of our work is that it opens entirely novel research perspectives. We are unaware of previous studies on the role of TEs in the human thymus. The drawback is that, as with all novel multi-omic systems biology studies, our work provides a roadmap for a multitude of future mechanistic studies that could not be realized at this stage. Indeed, we performed wet lab experiments to validate some but not all conclusions: (i) presentation of TE-derived MAPs by TECs and (ii) formation of dsRNAs in thymic pDCs. In response to Reviewer #1, we performed supplementary analyses to increase the robustness of our conclusions. Also, we indicated when conclusions relied strictly on correlative evidence and clarified the hypotheses drawn from our observations.

Regarding the Reviewer's questions about TE-derived dsRNAs, LINE, LTR, and SINE elements all have the potential to generate dsRNAs, given their highly repetitive nature and bi-directional transcription (1). As ~32% of TE subfamilies are overexpressed in pDCs, we hypothesized that these TE sequences might form dsRNA structures in these cells. To address the Reviewer's concerns regarding the amount of TE-derived RNAs among total cellular RNAs, we also computed the percentage of reads assigned to TEs in the different subsets of thymic APCs (see Reviewer 1 comment #4).

(2) Lack of generality of specific examples. This manuscript discusses the whole genomic picture of TE expression. In addition, one good way is to focus on the specific example to clearly discuss the biological significance of the acquisition of TEs for the thymic APC functions and the thymic selection.In figure 2, the authors focused on ETS-1 and its potential target genes ZNF26 and MTMR3, however, the significance of these genes in NK cell function or development is unclear. The authors should examine and discuss whether the distinct features of TEs can be found among the genomic loci that link to the fundamental function of the thymus, e.g., antigen processing/presentation.

We thank the Reviewer for this highly relevant comment. We investigated the genomic loci associated with NK cell biology to determine if ETS1 peaks would overlap with TE sequences in protein-coding genes' promoter region. Figure 2h illustrates two examples of ETS1 significant peaks overlapping TE sequences upstream of PRF1 and KLRD1. PRF1 is a protein implicated in NK cell cytotoxicity, whereas KLRD1 (CD94) dimerizes with NKG2 and regulates NK cell activation via interaction with the nonclassical MHC-I molecule HLA-E (2, 3). Thus, we modified the section of the manuscript addressing these results to include these new analyses:

"Finally, we analyzed publicly available ChIP-seq data of ETS1, an important TF for NK cell development (4), to confirm its ability to bind TE sequences. Indeed, 19% of ETS1 peaks overlap with TE sequences (Figure 2g). Notably, ETS1 peaks overlapped with TE sequences (Figure 2h, in red) in the promoter regions of PRF1 and KLRD1, two genes important for NK cells' effector functions (2, 3)."

(3) Since the deep analysis of the dataset yielded many intriguing suggestions, why not add a discussion of the biological reasons and significance?For example, in Figure 1, why is TE expression negatively correlated with proliferation? cTEC-TE is mostly postnatal, while mTEC-TE is more embryonic. What does this mean?

We thank the Reviewer for this comment. To our knowledge, the relationship between cell division and transcriptional activity of TEs has not been extensively studied in the literature. However, a recent study has shown that L1 expression is induced in senescent cells. We therefore added the following sentences to our Discussion:

"The negative correlation between TE expression and cell cycle scores in the thymus is coherent with recent data showing that transcriptional activity of L1s is increased in senescent cells (5). A potential rationale for this could be to prevent deleterious transposition events during DNA replication and cell division."

We also added several discussion points regarding the regulation of TEs by KZFPs to answer concerns raised by Reviewer 2 (see Reviewer 2 comment #1).

(4) To consolidate the experimental evidence about pDCs and TE-derived dsRNAs, one option is to show the amount of TE-derived RNA copies among total RNAs. The immunohistochemistry analysis in figure 4 requires additional data to demonstrate that overlapped staining was not caused by technical biases (e.g. uneven fixation may cause the non-specifically stained regions/cells). To show this, authors should have confirmed not only the positive stainings but also the negative staining (e.g. CD3, etc.). Another possible staining control was showing that non-pDC (CD303- cell fractions in this case) cells were less stained by the ds-RNA probe.

We thank the Reviewer for this suggestion. We computed the proportion of reads in each cell assigned to two groups of sequences known to generate dsRNAs: TEs and mitochondrial genes (1). These analyses showed that the proportion of reads assigned to TEs is higher in pDCs than other thymic APCs by several orders of magnitude (~20% of all reads). In contrast, reads derived from mitochondrial genes had a lower abundance in pDCs. We included these results in Figure 4 – figure supplement 2 and included the following text in the Results section entitled "TE expression in human pDCs is associated with dsRNA structures":

"To evaluate if these dsRNAs arise from TE sequences, we analyzed in thymic APC subsets the proportion of the transcriptome assigned to two groups of genomic sequences known as important sources of dsRNAs, TEs and mitochondrial genes (1). Strikingly, whereas the percentage of reads from mitochondrial genes was typically lower in pDCs than in other thymic APCs, the proportion of the transcriptome originating from TEs was higher in pDCs (~22%) by several orders of magnitude (Figure 4 – figure supplement 2)."

As a negative control for the immunofluorescence experiments, we used CD123- cells. Indeed, flow cytometry analysis of the magnetically enriched CD303+ fraction was around 90% pure, as revealed by double staining with CD123 and CD304 (two additional markers of pDCs): CD123- cells were also CD304-/lo, showing that these cells are non-pDCs. Thus, we decided to compare the dsRNA signal between CD123+ cells (pDCs) and CD123- cells (non-pDCs). The difference between CD123+ and CD123- cells was striking (Figure 4d).

**Reviewer #1 (Recommendations For The Authors):**
It was sometimes difficult for me to recognize the dot plots representing low expression against the white background. e.g., figure 1 supplement 1.

We thank the Reviewer for their comment, and we modified Figure 1 – figure supplement 1 as well as Figure 3 – figure 3 supplement 2 to improve the contrast between dots and background.

**Reviewer #2:**

**Reviewer #2 (Recommendations For The Authors):**
(1) In the abstract, results and discussion, the following conclusions are drawn that are not supported by the data: (a) TEs interact with multiple transcription factors in thymic cells, (b) TE expression leads to dsRNA formation, activation of RIG-I/MDA5 and secretion of IFN-alpha, (c) TEs are regulated by cell proliferation and expression of KZFPs in the thymus. All these statements derive from correlations. Only one TF has ChIP-seq data associated with it, dsRNA formation and/or IFN-alpha secretion could be independent of TE expression, and whilst KZFPs most likely regulate TEs in the thymus, the data do not demonstrate it. The authors also seem to suggests that AIRE, FEZF2 and CHD4 regulate TEs directly, but binding is not shown. The manuscript needs a thorough revision to be absolutely clear about the correlative nature of the described associations.

We agree with Reviewer #2 that some of the conclusions in our initial manuscript were not fully supported by experimental data. In the revised manuscript, we clearly indicated when conclusions relied strictly on correlative evidence and clarified the hypotheses drawn from our observations. Regarding the regulation of TE expression by AIRE, FEZF2, and CHD4, we reanalyzed publicly available ChIP-seq data of AIRE and FEZF2 in murine mTECs. For AIRE, we confirmed that ~30% of AIRE's statistically significant peaks overlap with TE sequences (see Reviewer 2, comment #6 for more details on read alignment and peak calling), confirming its ability to bind to TE sequences directly. We added these results to the main figures (Figure 5f) and modified the "AIRE, CHD4, and FEZF2 regulate distinct sets of TE sequences in murine mTECs" as follows:

“[…]. As a proof of concept, we validated that 31.42% of AIRE peaks overlap with TE sequences by reanalyzing ChIP-seq data, confirming AIRE's potential to bind TE sequences (Figure 5f)."

A reanalysis of FEZF2's ChIP-seq data yielded no significant peaks while using stringent criteria. For this reason, we decided to exclude these data and only use AIRE as a proof of concept.

Regarding KZFPs, we agree with Reviewer #2 that their impact on TE expression is probably significantly underestimated in our data. A potential reason for this is that KZFP expression is typically low; thus, transcriptomic signals from KZFPs could have been missed by the low depth of scRNA-seq. We mentioned this point in the Discussion:

"On the other hand, the contribution of KZFPs to TE regulation in the thymus is likely underestimated due to their typically low expression (6) and scRNA-seq's limit of detection."

(2) On the technical side, there are many dangers about analyzing RNA-seq data at the subfamily level and without stringent quality control checks. Outputs may be greatly confounded by pervasive transcription (see PMID 31425522), DNA contamination, and overlap of TEs with highly expressed genes. Whether TE transcripts are independent units or part of a gene also has important implications for the conclusions drawn. I would say that for most purposes of this work, an analysis restricted to independent TE transcripts, with appropriate controls for DNA contamination, would provide great reassurances that the results from subfamily-level analyses are sound. Showing examples from the genome browser throughout would also help.

We agree with the Reviewer that contamination could have interfered with TE quantification. We used FastQ Screen (7) to evaluate the contamination of our human scRNA-seq data. As illustrated in the Figure below, most reads aligned with the human genome, and there were no reads uniquely assigned to another species analyzed, confirming the high purity of our dataset.

**Author response image 2. sa3fig2:** 

As stated by the Reviewer, pervasive expression is another factor that can lead to overestimation of TE expression. To evaluate if pervasive expression impacted the results of our differential expression analysis of TEs between APC subsets, we visualized read alignment to TE sequences using a genome browser. We selected two samples containing the highest numbers of mTEC(II) and pDCs (T07_TH_EPCAM and FCAImmP7277556, respectively) and used STAR to align reads to the human genome (GRCh38). We then visualized read alignment to randomly selected loci of two subfamilies identified as overexpressed by mTEC(II) or pDCs (HERVE-int and Harlequin-int, respectively). The examples below show that the signal detected is specific to the TE sequences located in introns. Even though this visualization cannot guarantee that pervasive expression did not affect TE quantification in any way, it increases the confidence that the signal detected by our analyses genuinely originates from TE expression.

**Author response image 3. sa3fig3:** 

**Author response image 4. sa3fig4:** 

**Author response image 5. sa3fig5:** 

**Author response image 6. sa3fig6:** 

**Author response image 7. sa3fig7:** 

(3) Related to the above, it would be useful to describe in the main text (and methods) how multi-mapping reads are being handled. It wasn't clear to me how kallisto handles this, and it has implications for the results. In the analysis suggested above, only uniquely mapped reads would have to be used, despite its limitations.

We agree with the Reviewer that this information regarding assignment of multimapping reads is important. Kallisto uses an expectation-maximization (EM) algorithm to deal with multimapping reads, a strategy used by several algorithms developed to study TE expression (8). Briefly, the EM algorithm reassigns multimapping reads based on the number of uniquely mapped reads assigned to each sequence. Thus, we added the following details to the methods section:

"Preprocessing of the scRNA-seq data was performed with the kallisto (9), which uses an expectation-maximization algorithm to reassign multimapping reads based on the frequency of unique mappers at each sequence, and bustools workflow."

(4) Whilst I liked the basic idea, I am not convinced that correlating TE and TF expression is a good strategy for identifying TE-TF associations at enhancers. Enhancers express very low levels of short transcripts, which I doubt would be detected in low-depth scRNA-seq data. The transcripts the authors are using to make such associations may therefore have nothing to do with the enhancer roles of TEs. I would limit these analyses to cell types for which there is histone modification data and correlate TF expression with that instead.

We agree with the Reviewer that it would have been interesting to correlate the expression of TFs with signals of histone marks at TE sequences. However, we could not perform this analysis because we did not have matched data of histone marks throughout thymic development. Therefore, we adopted an alternative, well-suited strategy.

Our strategy to identify TE enhancer candidates is depicted in Figure 2a: (i) correlation between the expression of the TF and the TE subfamily, (ii) presence of the TF binding motif in the sequence of the TE enhancer candidate, and (iii) colocalization of the TE enhancer candidate with significant peaks of H3K27ac and H3K4me3 in the same cell type from the ENCODE Consortium ChIP-seq data. We limited our analyses to the eight cell types present both in our dataset and the ENCODE Consortium: B cells, CD4 Single Positive T cells (CD4 SP), CD8 Single Positive T cells (CD8 SP), dendritic cells (DC), monocytes and macrophages (Mono/Macro), NK cells, Th17, and Treg.

(5) Figure 2G: binding of ETS1 is unconvincing. Were there statistically meaningful peaks called in these regions? It would be good to also show a metaplot/heatmap of ETS1 profile over all elements of relevant subfamilies. Showing histone marks on the genome browser snapshots would also be useful. Is there any transcriptional evidence that the specific Alus shown act as alternative promoters?

We agree with the Reviewer that the examples provided were not particularly convincing. Thus, we reanalyzed the data to determine if statistically significant ETS1 peaks (see the answer to Reviewer 2's comment #6 for details on the methods) located near gene transcription start sites overlapped with TEs. We thereby provided examples of significant ETS1 peaks overlapping TE sequences in the promoter region of two prototypical NK cell protein-coding genes (Figure 2h).

(6) Why was -k 10 used with bowtie2? This will map the same read to multiple locations in the genome, increasing read density at more repetitive (younger) TEs. The authors should use either default settings, being clear about the outcome (random assignment of multimapping reads to one location), or use only uniquely aligned reads.

We thank the Reviewer for their comment and agree that using the -k 10 parameter with bowtie2 was not optimal for TE analysis. To improve the strength of our analyses, we reanalyzed all ChIP-seq data of our manuscript (Figure 2g and h, Figure 5e and f) using the following strategy: alignment with bowtie2 using default parameters except –very-sensitive, multimapping read removal with samtools view -q 10, removal of duplicate reads with samtools markdup -r, peaks calling was performed with macs2 with the -m 5 50 parameter, and peaks overlapping ENCODE's blacklist regions were removed with bedtools intersect.

These new analyses strengthen our evidence that TEs interact with multiple genes that regulate thymic development and function. We updated the results sections concerning ChIP-seq data analyses and the Methods section to include this information:

"ChIP-seq reads were aligned to the reference *Homo sapiens* genome (GRCh38) using bowtie2 (version 2.3.5) (10) with the --very-sensitive parameter. Multimapping reads were removed using the samtools view function with the -q 10 parameter, and duplicate reads were removed using the samtools markdup function with the -r parameter (11). Peak calling was performed with macs2 with the -m 5 50 parameter (12). Peaks overlapping with the ENCODE blacklist regions (13) were removed with bedtools intersect (14) with default parameters. Overlap of ETS1 peaks with TE sequences was determined using bedtools intersect with default parameters. BigWig files were generated using the bamCoverage function of deeptools2 (15), and genomic tracks were visualized in the USCS Genome Browser (16)."

(7) Figure 1d needs a y axis scale. Could the authors also provide details of how the random distribution of TE expression was generated?

We agree that the Reviewer that Figure 1d was incomplete and made the appropriate modifications. Regarding the random distribution, we reproduced our dataset containing the expression of 809 TE subfamilies in 18 cell populations. For each combination of TE subfamily and cell type, we randomly assigned an "expression pattern" as identified by the hierarchical clustering of Figure 1b. Then, we computed the maximal occurrence of an expression pattern across cell types for each TE subfamily to generate the distribution curve in Figure 1d. We added the following details to the Methods section to clarify how the random distribution was generated:

"As a control, a random distribution of the expression of 809 TE subfamilies in 18 cell populations was generated. A cluster (cluster 1, 2, or 3) was randomly attributed for each combination of TE subfamily and cell type, and the maximal occurrence of a given cluster across cell types was then computed for each TE subfamily. Finally, the distributions of LINE, LTR, and SINE elements were compared to the random distribution with Kolmogorov-Smirnov tests."

(8) The motif analysis requires a minimum of 1 locus from each TE subfamily containing it in order to be reported, but this seems like a really low threshold that will output a lot of noise. What is the rationale here?

We agree with the Reviewer that this threshold might appear low. Nonetheless, these analyses ultimately aimed to identify TE promoter and enhancer candidates. Hence, we did not want to put an arbitrary threshold at a higher value (e.g., a certain number or percentage of all loci of a given TE subfamily), as this might create a bias based on the total number of loci of a given TE subfamily. Moreover, our rationale was that a TE locus might act as a promoter/enhancer even if it is the only locus of its subfamily containing a TF binding site.

Even though this strategy might have created some noise in the analyses of interactions between TFs and TEs of Figure 2 (panels a-e), we are confident that our bootstrap strategy efficiently removed low-quality identifications based on low correlations values or expression of TF and TE in low percentages of cells. Additionally, the subsequent analyses on TE promoter and enhancer candidates were performed exclusively for the TE loci containing TF binding sites to avoid adding noise to these analyses.

(9) Figure 4e: is this a log2 enrichment? If not, the enrichments for some of the gene sets are not so high.

The enrichment values represented in Figure 4e are not log-transformed. It is essential to highlight that gene set enrichment values were computed for each possible pair of thymic APCs (e.g., pDC vs. cDC1, pDC vs. mTEC(II), etc.), and the values represented in Figure 4e are an average of each comparison pictured at the bottom of the UpSet plot.

However, we agree with Reviewer 2 that the average enrichment value is not extremely high. We thus made the following modifications to the Results section ("TE expression in human pDCs is associated with dsRNA structures") to better represent it:

"Notably, thymic pDCs harbored moderate yet significant enrichment of gene signatures of RIG-I and MDA5-mediated IFN ɑ/β signaling compared to all other thymic APCs (Figure 4e and Supplementary file 1 – Table 8)."

(10) Please be clear on results subtitles when these refer to mouse.

We apologize for the confusion and modified the subtitles to clarify if the results refer to mouse or human data.

(11) Figure 1 - figure supplement 2: "assignation" should be 'assignment'.

We thank the Reviewer for their keen eye and changed the title of Figure 1 – figure supplement 2.

(1) Sadeq S, Al-Hashimi S, Cusack CM, Werner A. Endogenous Double-Stranded RNA. Noncoding RNA. 2021;7(1).

(2) Kim N, Kim M, Yun S, Doh J, Greenberg PD, Kim TD, et al. MicroRNA-150 regulates the cytotoxicity of natural killers by targeting perforin-1. J Allergy Clin Immunol. 2014;134(1):195-203.

(3) Gunturi A, Berg RE, Forman J. The role of CD94/NKG2 in innate and adaptive immunity. Immunol Res. 2004;30(1):29-34.

(4) Taveirne S, Wahlen S, Van Loocke W, Kiekens L, Persyn E, Van Ammel E, et al. The transcription factor ETS1 is an important regulator of human NK cell development and terminal differentiation. Blood. 2020;136(3):288-98.

(5) De Cecco M, Ito T, Petrashen AP, Elias AE, Skvir NJ, Criscione SW, et al. L1 drives IFN in senescent cells and promotes age-associated inflammation. Nature. 2019;566(7742):73-8.

(6) Huntley S, Baggott DM, Hamilton AT, Tran-Gyamfi M, Yang S, Kim J, et al. A comprehensive catalog of human KRAB-associated zinc finger genes: insights into the evolutionary history of a large family of transcriptional repressors. Genome Res. 2006;16(5):669-77.

(7) Wingett SW, Andrews S. FastQ Screen: A tool for multi-genome mapping and quality control. F1000Res. 2018;7:1338.

(8) Lanciano S, Cristofari G. Measuring and interpreting transposable element expression. Nat Rev Genet. 2020;21(12):721-36.

(9) Bray NL, Pimentel H, Melsted P, Pachter L. Near-optimal probabilistic RNA-seq quantification. Nat Biotechnol. 2016;34(5):525-7.

(10) Langmead B, Salzberg SL. Fast gapped-read alignment with Bowtie 2. Nat Methods. 2012;9(4):357-9.

(11) Danecek P, Bonfield JK, Liddle J, Marshall J, Ohan V, Pollard MO, et al. Twelve years of SAMtools and BCFtools. Gigascience. 2021;10(2).

(12) Zhang Y, Liu T, Meyer CA, Eeckhoute J, Johnson DS, Bernstein BE, et al. Model-based analysis of ChIP-Seq (MACS). Genome Biol. 2008;9(9):R137.

(13) Amemiya HM, Kundaje A, Boyle AP. The ENCODE Blacklist: Identification of Problematic Regions of the Genome. Sci Rep. 2019;9(1):9354.

(14) Quinlan AR, Hall IM. BEDTools: a flexible suite of utilities for comparing genomic features. Bioinformatics. 2010;26(6):841-2.

(15) Ramirez F, Ryan DP, Gruning B, Bhardwaj V, Kilpert F, Richter AS, et al. deepTools2: a next generation web server for deep-sequencing data analysis. Nucleic Acids Res. 2016;44(W1):W160-5.

(16) Kent WJ, Sugnet CW, Furey TS, Roskin KM, Pringle TH, Zahler AM, et al. The human genome browser at UCSC. Genome Res. 2002;12(6):996-1006.